# Heavy-tailed Physics-Informed Neural Networks

Jephte Abijuru [1]   Mayank Nagda [1]   Jan Tauberschmidt [1,2]   Phil Sidney Ostheimer [1]
Sebastian Josef Vollmer [1,2]   Stephan Mandt [3]   Marius Kloft [1]   Sophie Fellenz [1]

## Abstract

Physics-informed neural networks (PINNs) enforce physical laws by minimizing partial differential equation (PDE) residuals and auxiliary constraints. Standard training relies on a mean-squared error (MSE) objective, which implicitly assumes independent Gaussian residuals with a fixed global variance. We show theoretically and empirically that residuals encountered during PINN training are heterogeneous and heavy-tailed, revealing a systematic mismatch with this assumption. As a consequence, a small number of large residuals can disproportionately dominate both the loss and gradient, leading to poorly balanced optimization dynamics. Motivated by this mismatch, we adopt a Student-$t$ residual model to explicitly capture heavy-tailed behavior. An equivalent hierarchical representation yields an expectation–maximization (EM) algorithm that alternates between estimating residual-dependent weights and optimizing network parameters via a weighted MSE objective, allowing existing PINN solvers to be reused in the M-step. The resulting training dynamics bound the influence of extreme residuals and admit almost sure convergence guarantees under standard stochastic optimization assumptions. Experiments across a diverse suite of challenging PDE benchmarks demonstrate consistently improved solution accuracy and robustness compared to standard PINN training.

## 1. Introduction

Physics-informed neural networks (PINNs) solve partial differential equations (PDEs) by enforcing governing equations, boundary conditions, and auxiliary constraints in the loss function. By embedding physical laws directly into neural network training, PINNs provide mesh-free solvers for forward and inverse problems and have been applied in domains such as fluid dynamics, heat transfer, materials science, geophysics, and biomedical modeling (Raissi et al., 2019; 2020; Almajid & Abu-Al-Saud, 2022).

Despite their conceptual appeal, PINNs are often difficult to train reliably. In practice, they may converge slowly or fail on nonlinear or multiscale PDEs (Krishnapriyan et al., 2021). Numerous approaches have been proposed to mitigate these failure modes, including architectural modifications (Zhao et al., 2024; Xu et al., 2025a; Wang et al., 2024), adaptive sampling strategies (Daw et al., 2023), and loss reweighting schemes and optimizers (Chen et al., 2025; Wang et al., 2021; 2022b; 2025; Xu et al., 2025b).

PINN training typically relies on mean–squared error (MSE) losses to enforce PDE constraints. This choice implicitly specifies a statistical model for the residuals: MSE training corresponds to maximum likelihood estimation under a zero-mean Gaussian residual model with fixed global variance, thereby enforcing homogeneous confidence across space and time.

We show theoretically and empirically that residuals arising during PINN training systematically deviate from this assumption. For nonlinear differential operators, residuals are highly uneven, state-dependent, and frequently heavy-tailed, reflecting localized stiffness, sharp solution features, and multiscale interactions.

This discrepancy between the Gaussian residual model implicit in MSE-based PINN training and the empirical structure of PDE residuals reflects a mis-specification of the residual model underlying standard PINN training and explains common training pathologies, including poorly balanced gradients, slow convergence, and sensitivity to loss weighting.

To address this mismatch at the modeling level, we adopt a Student-$t$ residual model that explicitly captures heavy-tailed and heterogeneous PDE residuals. This choice directly reflects the empirical structure of residuals induced by nonlinear operators, localized stiffness, and multiscale interactions.

[1]Department of Computer Science, RPTU University Kaiserslautern-Landau [2]DSA, German Research Center for AI [3]University of California, Irvine. Correspondence to: Jephte Abijuru <abijuru@rptu.de>.

*Proceedings of the $43^{rd}$ International Conference on Machine Learning*, Seoul, South Korea. PMLR 306, 2026. Copyright 2026 by the author(s).

Crucially, the Student-$t$ model admits an equivalent hierarchical representation in which residual variance is treated as an uncertain latent quantity with a conjugate prior. This representation yields an efficient expectation–maximization (EM) algorithm that alternates between estimating residual-dependent weights and optimizing network parameters via a weighted MSE objective. As a result, the proposed approach can be integrated seamlessly into existing PINN solvers, preserving their structure while correcting the statistical assumptions underlying residual enforcement.

Our contributions are summarized as follows:

- We provide a probabilistic interpretation of standard PINN training, showing that MSE-based objectives implicitly assume a fixed Gaussian model for PDE residuals.

- We demonstrate, through theory and empirical analysis, that residuals arising during PINN training for nonlinear PDEs are systematically heterogeneous and heavy-tailed.

- We introduce a Student-$t$ residual model for PINNs that captures this behavior and derive an efficient EM-based training procedure based on a hierarchical likelihood formulation.

- We demonstrate consistent accuracy and robustness gains across stiff and multiscale PDE benchmarks, achieving up to an order-of-magnitude reduction in relative $\ell_2$ error on classical PINN failure modes such as the Allen–Cahn and Wave Equations.

## 2. Rethinking the PINN Objective

In this section, we show that the standard PINN training objective admits an explicit probabilistic interpretation: it corresponds to maximum likelihood estimation under a fixed Gaussian model for the residual distribution. We then present empirical evidence that this assumption is systematically violated during training. Consider a partial differential equation with differential, initial, and boundary operators $\mathcal{P}, \mathcal{I}$, and $\mathcal{B}$. Let $u^*(x, t)$ denote the true solution satisfying $\mathcal{P}[u^*] = 0$, $\mathcal{I}[u^*] = 0$, and $\mathcal{B}[u^*] = 0$ on their respective domains. Approximating $u^*$ with a neural surrogate $u_\theta(x, t)$ induces residual functions

$$r_{\text{PDE}} = \mathcal{P}[u_\theta], \qquad r_{\text{IC}} = \mathcal{I}[u_\theta], \qquad r_{\text{BC}} = \mathcal{B}[u_\theta],$$

and, when data are available, $r_{\text{data}} = u_\theta - u_{\text{obs}}$.

In the standard PINN formulation, residuals are grouped by constraint type $g \in \mathcal{G} = \{\text{PDE}, \text{IC}, \text{BC}, \text{data}\}$. For each group one collects samples $\{r_i^{(g)}\}_{i=1}^{N_g}$ and assigns a positive

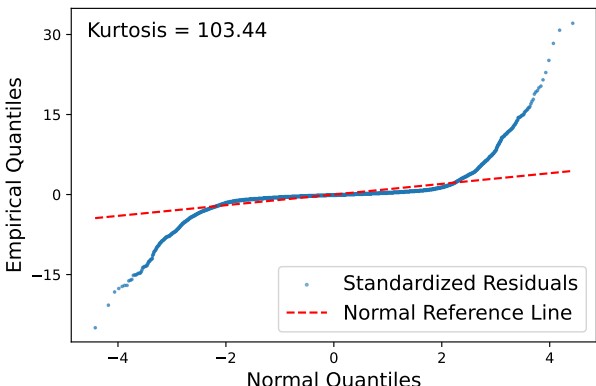

*Figure 1.* Normal Q–Q plot of standardized PDE residuals for the Burgers equation after training with a standard MSE-based PINN. Systematic deviations from the Gaussian reference line, particularly in the tails, indicate pronounced non-Gaussian behavior with extreme excess kurtosis (Kurtosis $\approx 103$).

weight $\lambda_g$, yielding the training objective

$$\mathcal{L}_{\text{PINN}}(\theta) = \sum_{g \in \mathcal{G}} \lambda_g \sum_{i=1}^{N_g} \left(r_i^{(g)}\right)^2. \tag{1}$$

Although commonly introduced as a penalty, the PINN objective can be interpreted probabilistically by specifying a distribution over residuals. For fixed network parameters $\theta$, sampling collocation points induces a real-valued random variable $r_\theta(x, t)$ whose realizations are the residual values entering the objective. Suppose that each residual $r_i^{(g)}$, conditioned on its sample $(x_i, t_i)$, is independently drawn from a zero-mean Gaussian distribution with group-specific variance $\sigma_g^2$,

$$p\left(r_i^{(g)} \mid x_i, t_i; \theta\right) = \mathcal{N}\left(0, \sigma_g^2\right).$$

Under this assumption, the total negative log-likelihood of all residuals, up to additive constants independent of $\theta$, reduces to

$$\mathcal{L}(\theta) = \sum_{g \in \mathcal{G}} \lambda_g \sum_{i=1}^{N_g} \left(r_i^{(g)}\right)^2, \qquad \lambda_g := \frac{1}{2\sigma_g^2}.$$

Thus, standard MSE-based PINN training corresponds to maximum likelihood estimation under a Gaussian residual model. Gaussianity imposes a restrictive structural constraint on residuals induced by nonlinear differential operators. Empirical evidence indicates systematic deviations from Gaussian behavior in residual distributions during training (Figure 1). We formalize this observation in the following hypothesis.

**Hypothesis 1** (Residual distribution hypothesis). *During PINN training, residual samples follow a zero-centered distribution that is heavier-tailed than a Gaussian.*

To quantify departures from Gaussian behavior, we use kurtosis as a diagnostic measure of tail heaviness. For a real-valued random variable $Z$ with finite mean $\mu$ and variance $\sigma^2$, the kurtosis is defined as

$$\operatorname{kurt}(Z) := \frac{\mathbb{E}[(Z-\mu)^4]}{\sigma^4}.$$

A Gaussian random variable satisfies $\operatorname{kurt}(Z) = 3$. Following prior work, we use kurtosis as an empirical diagnostic to characterize non-Gaussian and heavy-tailed residual behavior during PINN training (Daw et al., 2023; Cohen et al., 2020).

Using kurtosis as a diagnostic, we observe pronounced deviations from Gaussian residual behavior during PINN training. Figure 1 illustrates this effect for the Burgers equation, where standardized PDE residuals exhibit strong tail deviations from Gaussianity and extreme excess kurtosis. Additional qualitative analyses across further PDE benchmarks, including analogous Q–Q diagnostics, are provided in Appendix E.

Complementary theoretical results in Appendix F, in particular Theorem F.5, demonstrate that even near initialization, nonlinear PDE operators induce super-Gaussian residual distributions through multiplicative interactions of Gaussian network outputs and derivatives.

Taken together, these findings expose a systematic mismatch between the fixed Gaussian residual model implicit in MSE-based PINN training and the empirical structure of residuals encountered in practice. In the following section, we address this mismatch by introducing a learnable residual likelihood and deriving a corresponding training procedure.

## 3. Related Work

We situate our work within the literature on physics-informed neural networks by reviewing existing training strategies, robust residual treatments, and probabilistic formulations related to residual modeling.

**Physics-Informed Neural Networks (PINNs).** PINNs are a mesh-free framework for solving PDEs across many domains (Raissi et al., 2019; 2020), but they can be hard to train in complex regimes due to gradient stiffness and convergence to physically incorrect local minima (Wang et al., 2021; Krishnapriyan et al., 2021). Proposed fixes include new architectures (e.g., PINNsFormer) (Zhao et al., 2024), error-focused sampling (e.g., R3) (Daw et al., 2023), and improved optimization/loss balancing such as adaptive reweighting and alternative $L^p$ objectives (Wang et al.,

2022b;a). However, most approaches still rely on MSE's implicit Gaussian assumptions, which may conflict with the heavy-tailed residuals often observed in practice.

**Neural Operators and PINNs.** Neural operators such as DeepONet (Lu et al., 2021) and Fourier neural operators (Li et al., 2023) learn mappings between function spaces from labeled solution data and do not enforce governing equations during training. In contrast, PINNs incorporate physical constraints explicitly through residual penalties evaluated at sampled space–time locations. Because our work studies the statistical structure of residuals induced by physics-constrained training, the proposed heavy-tailed residual modeling is specific to PINNs and does not directly apply to data-driven operator learning. We therefore focus on comparisons within the PINN framework, where residual distributions arise as a direct consequence of enforcing physical laws.

**Robust Regression, IRLS, and EM.** Our formulation is closely related to classical robust regression, where heavy-tailed residual models are used to reduce sensitivity to outliers and departures from Gaussian noise assumptions (Lange et al., 1989; Huber, 1992; Peel & McLachlan, 2000). In particular, Student-$t$ regression has a standard Gaussian–Gamma scale-mixture representation, under which maximum-likelihood estimation can be carried out by expectation–maximization. The resulting updates are equivalent to an iteratively reweighted least-squares procedure in which large residuals receive smaller effective weights (Lange et al., 1989). Our contribution builds on the classic EM/IRLS machinery to bring this robust-regression perspective to PINN training, where the residuals are not supervised observation errors but PDE residuals produced by differential operators. Particularly, the role of the latent weights is not merely to downweight noisy labels or statistical observational outliers, but adaptively modulate the enforcement of physical constraints across space–time. To our knowledge, prior PINN work has not explicitly formalized PDE residuals as distributions to be modeled during training. Our work takes this perspective and models their empirically observed heavy-tailedness through a Student-$t$ residual likelihood, yielding a classical EM/IRLS-based weighting scheme for physics-constrained optimization.

**Loss Functions and Robust Training.** Several works revisit the mean-squared error objective in PINNs. For example, Wang et al. (2022a) show that quadratic penalties can fail in Hamilton–Jacobi–Bellman problems and propose alternative objectives based on $L^p$ norms. However, as noted by Daw et al. (2023), non-quadratic penalties can introduce undesirable training dynamics, including oscillatory behavior across competing residual modes. Relatedly, Daw et al. (2023) use kurtosis as a diagnostic to motivate adaptive

sampling strategies, interpreting heavy-tailed residuals as a symptom of localized training failure. While heavy-tailed behavior and heterogeneity are widely studied in deep learning (Kim et al., 2024; Ament et al., 2024; Pandey et al., 2025; Wong-Toi et al., 2024), they have not been systematically incorporated into PINN loss functions. In contrast, we define PINN training through an explicit Student-*t* residual likelihood, directly modeling heavy-tailed behavior rather than treating it as a secondary diagnostic or heuristic.

## 4. Residual Likelihood Learning

Section 2 motivates treating PINN training as likelihood-based inference on residuals. Under this framework, the choice of loss function corresponds to specifying a parametric form for the residual distribution and jointly inferring its parameters together with the network parameters $\theta$. In this section, we instantiate this framework using a Student-*t* residual model, a canonical heavy-tailed distribution that generalizes the Gaussian and is well suited to heterogeneous residual behavior observed empirically. We show that this choice admits a simple hierarchical representation and leads to an efficient EM procedure for joint inference of $\theta$ and the parameters of the residual likelihood.

Let $r_i \equiv r_\theta(x_i, t_i)$ denote a generic residual. For notational clarity, subsequent derivations suppress the residual group index $g$ and apply identically to each group. We instantiate the likelihood using a Student-*t* distribution,

$$r_i \overset{\text{i.i.d.}}{\sim} T(0, \lambda, \nu), \quad (2)$$

where $\lambda > 0$ is a global precision parameter and $\nu > 0$ controls tail heaviness. This distribution assigns high probability mass to small residuals while allowing occasional large deviations. As $\nu \to \infty$, the Student-*t* distribution converges to a Gaussian, recovering the MSE assumption as a limiting case.

The negative log-likelihood $-\log p(\mathbf{r} \mid \theta, \lambda, \nu)$ is, up to additive constants independent of $\theta$,

$$\mathcal{L}_{\mathrm{T}}(\theta; \lambda, \nu) = \sum_{i=1}^{N} \frac{\nu + 1}{2} \log\left(1 + \frac{\lambda \, r_i(\theta)^2}{\nu}\right). \quad (3)$$

For finite $\nu$, this objective grows sub-quadratically in the residuals and limits the influence of large violations, while in the limit $\nu \to \infty$ it reduces to the standard quadratic penalty.

Direct optimization of $\mathcal{L}_{\mathrm{T}}$ is difficult for nonlinear residuals due to its non-quadratic form. To obtain a tractable optimization scheme, we exploit the Gaussian–Gamma hierarchical representation of the Student-*t* distribution,

$$r_i \mid \eta_i, \lambda \sim \mathcal{N}\big(0, (\eta_i \lambda)^{-1}\big), \qquad \eta_i \mid \nu \sim \mathrm{Gam}\big(\tfrac{\nu}{2}, \tfrac{\nu}{2}\big), \quad (4)$$

where $\eta_i$ is a latent local precision variable. Marginalizing $\eta_i$ recovers the Student-*t* likelihood underlying (3).

Introducing a variational distribution $q(\boldsymbol{\eta})$, the log joint objective $\log p(\mathbf{r}, \theta, \lambda, \nu)$ admits the decomposition

$$\log p(\mathbf{r}, \theta, \lambda, \nu) = \mathcal{L}_{\mathrm{ELBO}}(q, \theta, \lambda, \nu) + \mathrm{KL}(q(\boldsymbol{\eta}) \,\|\, p(\boldsymbol{\eta} \mid \mathbf{r}, \theta, \lambda, \nu)), \quad (5)$$

where the evidence lower bound (ELBO) is defined as

$$\mathcal{L}_{\mathrm{ELBO}}(q, \theta, \lambda, \nu) := \mathbb{E}_{q(\boldsymbol{\eta})}[\log p(\mathbf{r}, \boldsymbol{\eta}, \theta | \lambda, \nu)] - \mathbb{E}_{q(\boldsymbol{\eta})}[\log q(\boldsymbol{\eta})]. \quad (6)$$

Choosing $q(\boldsymbol{\eta}) = p(\boldsymbol{\eta} \mid \mathbf{r}, \theta, \lambda, \nu)$ makes the bound tight (the KL term vanishes), and maximizing the ELBO is equivalent to maximizing the marginal Student-*t* objective with priors. Coordinate ascent on the ELBO therefore yields an EM algorithm whose stationary points coincide with those of $\mathcal{L}_{\mathrm{T}}$.

At iteration $k$, EM alternates between the following optimization problems:

**E-step:**

$$q^{(k+1)} = \arg\max_q \mathcal{L}_{\mathrm{ELBO}}\left(q, \theta^{(k)}, \lambda^{(k)}, \nu^{(k)}\right) \quad (7)$$

**M-step:**

$$(\theta^{(k+1)}, \lambda^{(k+1)}, \nu^{(k+1)}) = \arg\max_{\theta, \lambda, \nu} \mathcal{L}_{\mathrm{ELBO}}\left(q^{(k+1)}, \theta, \lambda, \nu\right) \quad (8)$$

For the conjugate model (4), the E-step is available in closed form. In particular, the posterior expectation of each latent precision is

$$w_i := \mathbb{E}[\eta_i \mid r_i, \theta, \lambda, \nu] = \frac{\nu + 1}{\nu + \lambda r_i(\theta)^2}. \quad (9)$$

These weights adapt to the magnitude of each residual, downweighting large deviations while leaving small residuals largely unaffected.

Holding $q(\boldsymbol{\eta})$ fixed, the M-step for $\theta$ reduces to the weighted least-squares problem

$$\theta^{(k+1)} = \arg\min_\theta \frac{\lambda}{2} \sum_{i=1}^{N} w_i \, r_i(\theta)^2 - \log p(\theta). \quad (10)$$

We place a Gamma prior on the precision parameter, $\lambda \sim \mathrm{Gam}(a_\lambda, b_\lambda)$, with shape $a_\lambda$ and rate $b_\lambda$. Maximizing the ELBO with respect to $\lambda$ then yields the closed-form update

$$\lambda^{(k+1)} = \frac{\frac{N}{2} + a_\lambda - 1}{b_\lambda + \frac{1}{2} \sum_{i=1}^{N} w_i r_i(\theta^{(k+1)})^2}. \quad (11)$$

**Algorithm 1** EM Training for Heavy-Tailed PINNs

1: **Input:** Samples $\{(x_i, t_i)\}_{i=1}^N$, initial parameters $\theta^{(0)}$, initial hyperparameters $(\lambda^{(0)}, \nu^{(0)})$, Gamma prior parameters $(a_\lambda, b_\lambda)$, prior $p(\nu)$
2: Initialize $\theta \leftarrow \theta^{(0)}, \lambda \leftarrow \lambda^{(0)}, \nu \leftarrow \nu^{(0)}$
3: **repeat**
4:      Evaluate residuals $r_i = r_i(\theta)$ for $i = 1, \ldots, N$
5:      **E-step:** update posterior expectations $w_i$ of latent precisions $\eta_i$ using (9)
6:      **M-step:** update model parameters given $\{w_i\}$
7:      (i) update network parameters $\theta$ by solving the weighted least-squares problem (10) using gradient-based optimization
8:      (ii) update the precision parameter $\lambda$ using the closed-form expression (11)
9:      (iii) update the degrees of freedom parameter $\nu$ by maximizing the ELBO with respect to $\nu$ using one or two Newton iterations on $\zeta = \log \nu$
10: **until** $\mathcal{L}_{\text{ELBO}}$ converged
11: **Output:** final $(\theta, \lambda, \nu)$ and weights $\{w_i\}$

The degrees of freedom parameter $\nu$ is updated by maximizing the ELBO with respect to $\nu$, which is carried out using one or two Newton iterations on the reparameterization $\zeta = \log \nu$.

All updates follow directly from the ELBO and are summarized in Algorithm 1. Detailed derivations of the ELBO, the E-step posterior, and the M-step updates are deferred to Appendix C.

**Interpretation:** Algorithm 1 performs coordinate ascent on $\mathcal{L}_{\text{ELBO}}$ and can be interpreted as an iteratively reweighted least-squares procedure, with weights given by exact posterior expectations under the Student-$t$ residual model.

*Remark* 4.1. On priors: From a MAP perspective, explicit parameter priors $\log p(\theta)$ correspond to regularization terms in the optimization objective; in deep neural networks, such regularization is often implicit, arising from architectural inductive biases and stochastic optimization (Bishop, 2006; Mandt et al., 2017). Moreover, if the degrees-of-freedom parameter $\nu$ in Algorithm 1 is assigned a flat prior $p(\nu) \propto 1$, its value is determined entirely by the variational objective.

## 5. Training Dynamics

To connect the probabilistic modeling choice introduced earlier to optimization behavior, we analyze the training dynamics induced by the Student-$t$ negative log-likelihood when used to train PINNs. Our aim is to establish that this loss satisfies the standard assumptions required by stochastic approximation theory, thereby ensuring well-behaved stochastic gradient updates even in the presence of heavy-tailed residuals.

Let $u_\theta(x, t)$ denote the neural surrogate and let $r_\theta(x, t)$ denote the corresponding PDE residual evaluated at a collocation point $(x, t)$,

$$r_\theta(x, t) = \mathcal{P}[u_\theta](x, t),$$

where $\mathcal{P}$ is the (possibly nonlinear) differential operator defining the PDE. For notational simplicity, we write $r_i(\theta) \equiv r_\theta(x_i, t_i)$ for residuals evaluated at sampled collocation points $\{(x_i, t_i)\}$.

The Student-$t$ negative log-likelihood per residual is given by

$$\ell(r) = \frac{\nu + 1}{2} \log\left(1 + \frac{\lambda r^2}{\nu}\right), \qquad (12)$$

with degrees of freedom $\nu > 0$ and precision parameter $\lambda > 0$.

**Definition 5.1** (Score function). The score function associated with the Student-$t$ loss is defined as

$$\vartheta(r) := \ell'(r) = (\nu + 1)\frac{\lambda r}{\nu + \lambda r^2}.$$

For neural networks $u_\theta$ with smooth activation functions, the gradient of the loss with respect to the parameters can be written as

$$\nabla_\theta \ell(r_i(\theta)) = \vartheta(r_i(\theta)) \nabla_\theta r_\theta(x_i, t_i),$$

where $\nabla_\theta r_\theta(x_i, t_i)$ involves derivatives of $u_\theta$ up to the order prescribed by the operator $\mathcal{P}$. Thus, each residual influences the parameter update only through the scalar score function $\vartheta$.

**Proposition 5.2** (Bounded score). *For all $r \in \mathbb{R}$, the score function satisfies*

$$|\vartheta(r)| \leq \frac{\nu + 1}{2}\sqrt{\frac{\lambda}{\nu}}.$$

Proposition 5.2 (see Appendix D for a proof) shows that individual residuals contribute uniformly bounded terms to stochastic gradient updates, in contrast to MSE training where the corresponding contribution grows linearly with $|r|$. This boundedness ensures that the standard moment conditions required by stochastic approximation theory are satisfied, including in the presence of heavy-tailed residuals. Under standard smoothness and step-size conditions for stochastic gradient methods, stochastic gradient descent applied to the Student-$t$ objective converges almost surely to first-order stationary points. A formal convergence statement and proof are provided in Lemma D.1.

**Lemma 5.3** (Variational form of the Student-$t$ penalty). *Given $\nu, \lambda > 0$, the Student-t penalty defined in (12) admits a variational form*

$$\ell(r) = \min_{w>0} \left\{ \frac{\lambda}{2} w r^2 + c(w) \right\},$$

*with unique minimizer*

$$w(r) = \frac{\nu + 1}{\nu + \lambda r^2}$$

*and $c(w)$ indenpendent of $r$. Consequently, for any fixed $w > 0$, the right-hand side is a quadratic upper bound on $\ell(r)$, tight at $r$ when $w = w(r)$.*

**Theorem 5.4** (Quadratic majorization and descent). *Define $w_i^{(t)} := \frac{\nu+1}{\nu + \lambda r_i(\theta^{(t)})^2}$ and*

$$Q_t(\theta) := \frac{\lambda}{2} \sum_i w_i^{(t)} r_i(\theta)^2.$$

*Then, up to a constant independent of $\theta$, $\mathcal{L}_T(\theta) \le Q_t(\theta)$ for all $\theta$, with equality at $\theta = \theta^{(t)}$. Hence any generalized M-step satisfying $Q_t(\theta^{(t+1)}) \le Q_t(\theta^{(t)})$ implies $\mathcal{L}_T(\theta^{(t+1)}) \le \mathcal{L}_T(\theta^{(t)})$.*

This result shows that EM is a surrogate descent method: each iteration constructs a tight upper bound on the marginal objective and minimizes it with respect to $\theta$. In the Gaussian limit $\nu \to \infty$, the surrogate becomes exact and EM reduces to standard MSE training. Proofs of Lemma 5.3 and Theorem 5.4 are given in Appendix D.

## 6. Experiments

This section evaluates the proposed tail-aware PINN framework (t-PINN) through an empirical study on a wide range of PDE benchmarks characterized by failure modes. The experiments are organized to assess not only whether heavy-tailed residual modeling improves robustness, but also the consistency of these improvements across architectures and the underlying mechanisms responsible for them.

To this end we address the following questions:

1. Does replacing the Gaussian residual model used in standard PINNs with a heavy-tailed alternative improve robustness on established PINN failure-mode benchmarks?

2. Are any observed improvements consistent across network architectures?

3. To what extent do adaptive residual reweighting, heavy-tailed likelihood modeling, and EM-style optimization each contribute to robustness?

To address these questions, we evaluate t-PINN on a suite of PDE benchmarks chosen to expose known failure modes of MSE-based training, including sharp gradients, strong nonlinearities, multiscale dynamics, and higher-order differential operators. The benchmark suite spans wave propagation, nonlinear conservation laws, and reaction–diffusion systems. Detailed problem definitions, domains, initial and boundary conditions, and reference solutions are provided in Appendix A.

**Implementation of t-PINN** We implement t-PINN using the Algorithm 1 initialized with identical precision parameters $\lambda$ and degrees of freedom $\nu$, held fixed across t-PINN runs and architectures. We provide information on this initialization in Table 5. During training, the EM procedure is unrolled into a finite sequence of alternating E- and M-steps. At each E-step, residual-dependent weights are recomputed in closed form given the current model and likelihood parameters. The M-step consists of an inner loop of mini-batch gradient-based updates of the model parameters, implemented using Adam, followed by an update of the likelihood parameters $(\nu, \lambda)$.

**PINN Failure-Mode Benchmarks** We first evaluate t-PINN on established PINN failure-mode benchmarks established in prior works (Xu et al., 2025a;b), to assess whether heavy-tailed residual modeling improves robustness in regimes where MSE-based PINNs are known to break down.

For fair comparison, we report baseline results as published in Xu et al. (2025a;b), including methods based on architectural modifications (QRes (Bu & Karpatne, 2021), PINNs-Former (Zhao et al., 2024), RoPINNs (Wu et al., 2024), and PINNMamba (Xu et al., 2025a)) as well as numerical precision control (PINNFP64 (Xu et al., 2025b)). Except for PINNFP64, all other baselines and our model use single precision. These baselines represent the current state of the art on the considered failure-mode benchmarks. In this setting, we fix the backbone capacity and use a three-layer fully connected MLP with 32 neurons per layer for t-PINN to isolate the effect of the proposed training algorithm on known PINN failure modes; architectural scaling effects are examined in subsequent paragraphs. Complete PDE setups and implementation details are provided in Appendix B.

Table 1 reports results on four failure-mode benchmarks (Wave, Reaction, Convection, and Allen–Cahn). We report rRMSE (relative $\ell_2$ error) and rMAE (relative $\ell_1$ error) over the full space–time domain.

t-PINN achieves the best performance on all four benchmarks for both rRMSE and rMAE. Compared to the second-best method in Table 1 (the underlined entry), t-PINN improves rRMSE by 76.5% on Wave, 64.1% on Reaction,

*Table 1.* Comparison of t-PINN, which uses several orders of magnitude fewer parameters, with state-of-the-art methods on failure modes in PINNs. Metrics reported are rRMSE (relative $\ell_2$ error) and rMAE (relative $\ell_1$ error) over the full space–time domain. Underlined entries are the *second-best* results (best baseline) per metric/problem; Improvement (%) is computed w.r.t. the second-best result.

| Model | #Params | Wave | | Reaction | | Convection | | Allen–Cahn | |
|---|---|---|---|---|---|---|---|---|---|
| | | rRMSE | rMAE | rRMSE | rMAE | rRMSE | rMAE | rRMSE | rMAE |
| PINN-FP32 (JCP'19) | 527361 | 0.4141 | 0.2746 | 0.9785 | 0.9788 | 0.8989 | 0.6904 | 0.9404 | 0.9720 |
| QRes (ICDM'21) | 396545 | 0.5265 | 0.5335 | 0.9830 | 0.9826 | 0.9245 | 0.7498 | 0.8800 | 0.9821 |
| PINNsFormer (ICLR'24) | 453561 | 0.3632 | 0.3492 | 0.0296 | 0.0147 | 0.5217 | 0.0327 | 0.2236 | 0.9908 |
| RoPINNs (NeurIPS'24) | 527361 | 0.1720 | 0.0631 | 0.0170 | 0.0589 | 0.7200 | 0.6251 | – | – |
| PINNMamba (ICML'25) | 285763 | 0.0199 | 0.0193 | 0.0217 | 0.0092 | 0.0201 | 0.0184 | 0.0583 | 0.1432 |
| PINN-FP64 (NeurIPS'25) | 527361 | 0.0081 | 0.0080 | 0.0502 | 0.0271 | 0.0072 | 0.0059 | 0.0545 | 0.0157 |
| **t-PINN (ours)** | **3297** | **0.0019** | **0.0019** | **0.0061** | **0.0021** | **0.0059** | **0.0053** | **0.0097** | **0.0046** |
| Improvement (%) | – | 76.5 | 76.6 | 64.1 | 77.5 | 18.1 | 9.4 | 82.2 | 70.6 |

18.1% on Convection, and 82.2% on Allen–Cahn. The improvements in rMAE follow the same pattern: 76.6% on Wave, 77.5% on Reaction, 9.4% on Convection, and 70.6% on Allen–Cahn.

These results show that t-PINN performs well across different types of failure modes, including oscillatory solutions, reaction-dominated dynamics, convection-dominated transport, and phase-field dynamics. In addition, t-PINN uses only 3,297 parameters, while the strongest baselines use roughly 0.29 to 0.53 million parameters. This indicates that the gains come from the method itself rather than from increasing model size.

**Architecture-controlled Comparisons.** Are the observed gains tied to a specific architecture, or do they persist across PINN backbones? To this end, we compare standard MSE-based PINN training and t-PINN across three representative backbones: a standard multilayer perceptron (MLP), a modified MLP commonly used in PINN implementations (Wang et al., 2021), and PirateNet (Wang et al., 2024). Unless otherwise noted, all models employ the Tanh activation function. Training is carried out using mini-batch stochastic optimization with the Adam optimizer, where the set of collocation points is resampled at every iteration. The learning rate follows a two-stage schedule: a linear warm-up over the first 5000 steps, increasing from zero to $10^{-3}$, followed by exponential decay with a decay factor of 0.9. Adaptive loss balancing is applied for the modified MLP and PirateNet backbones, consistent with their original formulations (Wang et al., 2024). The standard MLP is trained without loss balancing to maintain a minimal and uniform baseline across experiments. For time-dependent PDEs, causal training is employed to mitigate violations of temporal causality.

For all experiments, to ensure a fair comparison, the total number of training iterations is matched to the baselines, with the outer loop executed every 1000 steps. For ease of

reproducibility, the hyperparameter settings are provided in Appendix B. The proposed loss is applied only to the PDE residuals evaluated in the interior of the domain, while boundary and initial condition constraints are enforced using a standard MSE loss.

Table 2 reports relative $\ell_2$ errors across a suite of PDEs, including the Wave, Burgers, Allen–Cahn, Korteweg–de Vries (KdV), and Ginzburg–Landau equations. All reported values are averaged over five random runs, following prior works. For the standard MLP, replacing the Gaussian residual model with the proposed heavy-tailed alternative yields substantial reductions in relative $\ell_2$ error, including approximately $5.6\times$ for the Wave equation, $3.2\times$ for Burgers, $7.8\times$ for Allen–Cahn, $5.5\times$ for KdV, and $2.2\times$ for Ginzburg–Landau. Moreover, these gains persist when the proposed loss is applied to deeper PINN backbones, indicating that t-PINN is compatible with existing PINN frameworks.

**Qualitative Behavior.** We next examine how heavy-tailed residual modeling alters the spatial and temporal structure of PINN solutions. As shown in Figure 2, MSE-trained PINNs tend to localize error near shocks, interfaces, and regions of rapid variation, where a small number of large residuals dominate gradients and distort optimization. In contrast, t-PINNs reduce the influence of extreme residuals, leading to a more even distribution of error in space and time. As a result, t-PINNs more accurately resolve regions with sharp gradients and interfaces while maintaining comparable accuracy elsewhere.

**Ablation on the Robustness of the EM Algorithm.** We ablate two aspects of the proposed t-PINN formulation: (i) the introduction of an E-step that assigns adaptive residual weights $w_i$, and (ii) Fixing or learning the degrees-of-freedom parameter $\nu$ of the Student-$t$ likelihood. The results are summarized in Table 3.

In the Gaussian baseline, the E-step is omitted and all resid-

*Table 2.* Comparison of PINN performance across network architectures and residual models. All models are trained using the same optimization pipeline. The evaluation metric is the relative $\ell_2$ error over the full space–time domain. Lower is better. Best results are shown in bold.

| Benchmark | MLP | | Modified MLP | | PirateNet | |
|---|---|---|---|---|---|---|
| | MSE-PINN | t-PINN | MSE-PINN | t-PINN | MSE-PINN | t-PINN |
| Wave | $9.0 \times 10^{-3}$ | $\mathbf{1.6 \times 10^{-3}}$ | $2.7 \times 10^{-4}$ | $\mathbf{2.0 \times 10^{-4}}$ | $7.1 \times 10^{-5}$ | $\mathbf{6.9 \times 10^{-5}}$ |
| Burgers | $1.1 \times 10^{-4}$ | $\mathbf{6.8 \times 10^{-5}}$ | $7.9 \times 10^{-5}$ | $\mathbf{4.6 \times 10^{-5}}$ | $5.4 \times 10^{-5}$ | $\mathbf{3.7 \times 10^{-5}}$ |
| Allen–Cahn | $2.1 \times 10^{-2}$ | $\mathbf{2.7 \times 10^{-3}}$ | $5.1 \times 10^{-5}$ | $\mathbf{2.1 \times 10^{-5}}$ | $\mathbf{1.4 \times 10^{-5}}$ | $1.9 \times 10^{-5}$ |
| Korteweg–de Vries | $1.6 \times 10^{-1}$ | $\mathbf{2.9 \times 10^{-2}}$ | $2.0 \times 10^{-3}$ | $\mathbf{4.6 \times 10^{-4}}$ | $\mathbf{4.3 \times 10^{-4}}$ | $5.1 \times 10^{-4}$ |
| Ginzburg–Landau | $1.9 \times 10^{-2}$ | $\mathbf{8.6 \times 10^{-3}}$ | $3.1 \times 10^{-2}$ | $\mathbf{9.9 \times 10^{-3}}$ | $1.49 \times 10^{-2}$ | $\mathbf{7.8 \times 10^{-3}}$ |

*Table 3.* Ablation of residual modeling on the one-dimensional convection equation. Moving from a Gaussian objective to a Student-*t* likelihood substantially reduces error, and learning the degrees of freedom $\nu$ yields an additional improvement.

| Model | $w_i$ | $\nu$ | Error ↓ |
|---|---|---|---|
| Gaussian baseline | 1 | – | 0.0304 (0.0200) |
| Student-*t*, fixed $\nu$ | EM | fixed | 0.0109 (0.0005) |
| Student-*t*, learned $\nu$ | EM | learned | **0.0028** (0.0009) |

uals are assigned unit weights $w_i \equiv 1$, resulting in an unweighted MSE objective in the M-step. Replacing the Gaussian objective with a Student-*t* likelihood activates the E-step of Algorithm 1, yielding residual-dependent weights and a weighted MSE in the M-step.

When the degrees-of-freedom parameter $\nu$ is fixed, weights adapt to residuals (in E-step) but with a fixed tail-heaviness schedule. Allowing $\nu$ to be updated in the M-step further adapts the weighting through successive EM iterations, leading to lower error. Across all settings, the precision parameter $\lambda$ is regularized by the Gamma prior introduced in Algorithm 1, whose parameters scale with the batch size.

Results are reported as mean $\pm$ standard deviation of relative $\ell^2$ error over five random initializations using the same MLP backbone. Additional ablations on hyperparameter initialization are provided in Appendix G.

## 7. Conclusion

We recast PINNs as likelihood-based models over constraint residuals, revealing standard MSE training as a Gaussian special case and motivating the learning of residual distributions. A Student-*t* instantiation yields a tractable inference procedure with improved robustness on challenging PDEs, while preserving the standard PINN framework. This perspective positions heavy-tailed residual likelihood learning as a foundation for PINNs and a starting point for more expressive residual models and inference schemes.

The residual likelihood learning mechanism developed here

naturally extends to more expressive models. We note that prior distributions in the EM algorithm introduce additional hyperparameters, and that optimizing all distributional parameters freely in the M-step may introduce numerical and statistical challenges. One direction is to consider non-parametric or implicit distributions that relax parametric assumptions on residual structure. In this work, we focus on likelihoods that admit tractible latent-variable representations and exact inference, which enables a clear probabilistic interpretation and theoretical analysis of the resulting optimization dynamics. Exploring more expressive residual models within this framework is an interesting direction for future work.

More broadly, our study reflects a recurring evaluation challenge in PINNs. Forward PDE solves are useful controlled benchmarks, but they are not necessarily the regimes where PINNs are most competitive with mature classical solvers. Future evaluations would benefit from benchmarks and evaluations that separately assess standard forward-solve performance and settings involving inverse problems, sparse observations, parameter identification, or differentiable optimization, where PINN-based methods may offer distinct advantages.

## Acknowledgements

The authors acknowledge support by the DFG through FOR 5359 (ID 459419731), TRR 375 (ID 511263698), SPP 2298 (IDs 441826958 and 441826958), and SPP 2331 (IDs 441958259, 553345933, and 466468799), by the Carl-Zeiss Foundation through the initiative AI-Care, and by the BMFTR award 01IS24071A. Stephan Mandt acknowledges funding from the National Science Foundation (NSF) through an NSF CAREER Award IIS-2047418, IIS2007719, the NSF LEAP Center, and the Hasso Plattner Research Center at UCI.

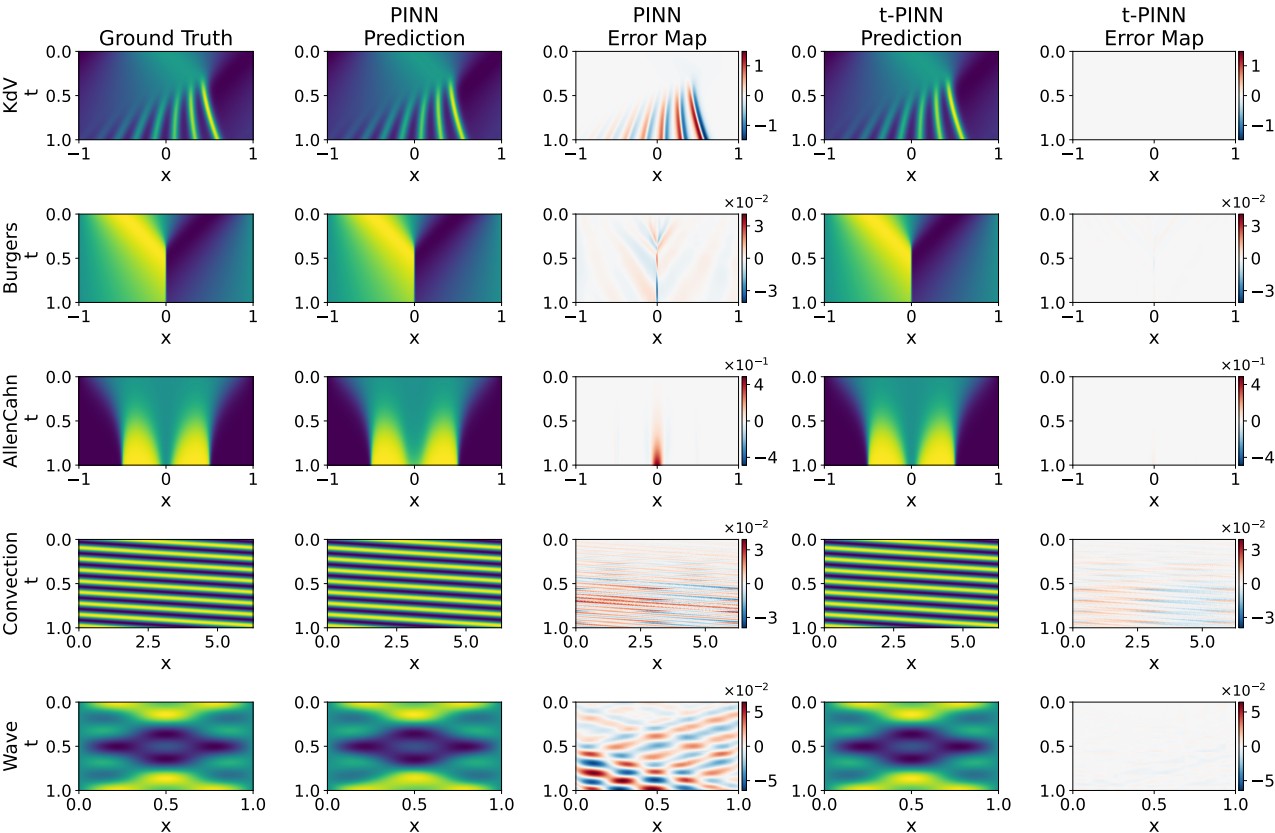

*Figure 2.* Qualitative comparison of a standard MSE-trained PINN and a t-PINN across multiple PDEs. The MSE-trained PINN exhibits localized regions of elevated error, particularly near sharp solution interfaces and regions of rapid variation, most notably for the KdV, Burgers, and Allen–Cahn equations. In contrast, the t-PINN maintains more uniform accuracy across these regions.

## Impact Statement

This paper presents work whose goal is to advance the field of machine learning, specifically physics-informed neural networks for solving partial differential equations. By introducing a heavy-tailed residual likelihood and an EM-based training procedure that remains compatible with standard PINN implementations, our approach aims to improve training stability and solution accuracy on challenging nonlinear, and multiscale PDEs.

These improvements may broaden the practical applicability of PINNs in scientific computing and engineering, including surrogate modeling, inverse problems, and data-efficient simulation where enforcing known physical structure is beneficial. We do not foresee direct negative societal impacts arising uniquely from this contribution beyond those commonly associated with general improvements in machine-learning-based modeling tools.

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

# A. PDE Setups and Metrics

## A.1. Metrics

In our experiments, we report the relative root mean squared error (rRMSE) and relative mean absolute error(rMAE) as the metric. For a set of evaluation points $\mathcal{S}$, model prediction $u_\theta$, and ground-truth solution $u^*$, we define

$$\text{rMAE} = \frac{\sum\limits_{x \in \mathcal{S}} \left| u_\theta(x) - u^*(x) \right|}{\sum\limits_{x \in \mathcal{S}} \left| u^*(x) \right|}, \tag{13}$$

$$\text{rRMSE} = \sqrt{\frac{\sum\limits_{x \in \mathcal{S}} \left( u_\theta(x) - u^*(x) \right)^2}{\sum\limits_{x \in \mathcal{S}} \left( u^*(x) \right)^2}}. \tag{14}$$

Note that both $u_\theta(x)$ and $u^*(x)$ can take positive or negative values; consequently, rMAE and rMSE may exceed 1.

## A.2. Benchmarks

To comprehensively test our algorithm, we include seven benchmarks. The first three correspond to canonical PDEs widely used in the PINN literature, while the last four correspond to non-linear high-dimensional baselines Wang et al. (2024).

**1D–Reaction.** This one-dimensional nonlinear ODE models chemical reactions:

$$\frac{\partial u}{\partial t} - \rho u(1 - u) = 0, \quad x \in (0, 2\pi), \ t \in (0, 1),$$

with initial and boundary conditions

$$u(x, 0) = \exp\left( -\frac{(x - \pi)^2}{2(\pi/4)^2} \right), \quad u(0, t) = u(2\pi, t).$$

The analytic solution is

$$u(x, t) = \frac{h(x)e^{\rho t}}{h(x)e^{\rho t} + 1 - h(x)}, \quad h(x) = \exp\left( -\frac{(x - \pi)^2}{2(\pi/4)^2} \right),$$

with $\rho = 5$. Prior work (Raissi et al., 2019; Krishnapriyan et al., 2021) identified this case as a "PINN failure mode" due to the nonlinear term, and its sharp interior boundary adds further difficulty. Following PINNsFormer (Xu et al., 2025a), we sample 101 points on the initial/boundary sets and a $101 \times 101$ grid on the residual domain. Evaluation uses the same mesh.

**1D–Wave.** A standard hyperbolic PDE from acoustics and fluid dynamics:

$$\frac{\partial^2 u}{\partial t^2} - 4\frac{\partial^2 u}{\partial x^2} = 0, \quad x \in (0, 1), \ t \in (0, 1),$$

with initial and boundary conditions

$$u(x, 0) = \sin(\pi x) + \tfrac{1}{2} \sin(\beta \pi x), \quad \frac{\partial u(x, 0)}{\partial t} = 0, \quad u(0, t) = u(1, t) = 0.$$

The analytic solution is

$$u(x, t) = \sin(\pi x) \cos(2\pi t) + \tfrac{1}{2} \sin(\beta \pi x) \cos(2\beta \pi t),$$

with $\beta = 3$. Compared to Reaction and Convection, the solution is smoother, making it easier for deep models. Training/evaluation meshes are sampled as in Reaction.

**1D–Convection.** A hyperbolic PDE relevant in fluids, atmosphere, and heat transfer:

$$\frac{\partial u}{\partial t} + \beta \frac{\partial u}{\partial x} = 0, \quad x \in (0, 2\pi), \ t \in (0, 1),$$

with

$$u(x, 0) = \sin(x), \quad u(0, t) = u(2\pi, t).$$

The analytic solution is $u(x, t) = \sin(x - \beta t)$, where we set $\beta = 50$. Despite its simple closed form, this problem is challenging for PINNs due to the high-frequency oscillations and sharp loss landscape (Krishnapriyan et al., 2021). Training/evaluation meshes follow the same setup as above.

**Allen–Cahn (AC).** A nonlinear reaction–diffusion equation that is widely used as a challenging benchmark for PINN models. The equation is given by

$$\frac{\partial u}{\partial t} - 0.0001 \frac{\partial^2 u}{\partial x^2} + 5u^3 - 5u = 0, \quad x \in [-1, 1], \ t \in [0, 1],$$

with initial and boundary conditions

$$u(x, 0) = x^2 \cos(\pi x),$$

$$u(-1, t) = u(1, t), \quad \frac{\partial u}{\partial x}(-1, t) = \frac{\partial u}{\partial x}(1, t).$$

Data generation follows Wang et al. (2024).

**Korteweg–de Vries (KdV).** A nonlinear dispersive PDE describing the evolution of solitary waves:

$$\frac{\partial u}{\partial t} + \eta u \frac{\partial u}{\partial x} + \mu^2 \frac{\partial^3 u}{\partial x^3} = 0, \quad x \in [-1, 1], \ t \in (0, 1),$$

with

$$u(x, 0) = \cos(\pi x), \quad u(-1, t) = u(1, t).$$

We use the standard parameter values $\eta = 1$ and $\mu = 0.022$. Data generation follows Wang et al. (2024).

**Ginzburg–Landau (GL).** We consider the two-dimensional complex Ginzburg–Landau equation

$$\frac{\partial A}{\partial t} = \varepsilon \Delta A + \mu A - \gamma A |A|^2, \quad (x, y) \in (-1, 1)^2, \ t \in (0, 1),$$

with initial condition

$$A(x, y, 0) = (10y + 10ix) \exp\left(-0.01(2500x^2 + 2500y^2)\right),$$

where $A$ is a complex-valued field and we set $\varepsilon = 0.004$, $\mu = 10$, and $\gamma = 10 + 15i$.

Writing $A = u + iv$ yields the real-valued system

$$\frac{\partial u}{\partial t} = \varepsilon \Delta u + \mu\left(u - (u - 1.5v)(u^2 + v^2)\right),$$

$$\frac{\partial v}{\partial t} = \varepsilon \Delta v + \mu\left(v - (v + 1.5u)(u^2 + v^2)\right).$$

Data generation follows Wang et al. (2024).

**Burgers.** A nonlinear convection–diffusion equation exhibiting shock formation and chaotic behavior:

$$\frac{\partial u}{\partial t} + \frac{1}{2} \frac{\partial}{\partial x}\left(u^2\right) - \nu \frac{\partial^2 u}{\partial x^2} = 0, \quad x \in [-1, 1], \ t \in (0, 1),$$

with initial and boundary conditions

$$u(x, 0) = -\sin(\pi x), \quad u(-1, t) = u(1, t).$$

We set the viscosity parameter to $\nu = 0.01/\pi$. Data generation follows Wang et al. (2024).

# B. Implementation Details

Beyond the Failure-Mode benchmark, we also test t-PINN against more complex PDEs detailed in A.2.

## B.1. Failure Modes Baselines

We evaluate **t-PINN** against a set of state-of-the-art physics-informed learning models that include original formulations, optimization-oriented refinements, architectural enhancements, and recent methods designed to better capture spatio-temporal structure. This selection enables a controlled comparison across differing modeling and training strategies.

- **Canonical PINN**: We include the original Physics-Informed Neural Network introduced by Raissi et al. (2019). This framework employs a Multilayer Perceptron (MLP) to approximate the solution $u(x, t)$ from spatio-temporal inputs $(x, t)$. Training minimizes a composite loss consisting of PDE residuals, initial conditions, and boundary conditions, with derivatives computed via automatic differentiation. Despite its generality, this formulation is prone to optimization pathologies and often exhibits inaccurate propagation of initial conditions, leading to overly smooth or physically inconsistent solutions.

- **PINN+FP64**: We compare against the high-precision PINN baseline proposed by Xu et al. (2025b), which performs training entirely in double precision (FP64). This approach targets numerical instability and gradient underflow commonly observed in standard FP32 PINNs. The architecture and loss remain unchanged, allowing this baseline to isolate the effect of numerical precision on optimization behavior.

- **QRes (Quadratic Residual Networks)**: QRes introduces quadratic residual connections into the standard MLP architecture to increase representational capacity for complex nonlinear dynamics (Bu & Karpatne, 2021). This modification aims to improve approximation quality for challenging PDEs while retaining the conventional PINN loss formulation.

- **RoPINN (Region Optimized PINN)**: RoPINN extends the standard PINN objective by optimizing PDE residuals over continuous neighborhood regions rather than isolated collocation points (Wu et al., 2024). This is achieved via Monte Carlo sampling within adaptively defined trust regions, enabling improved enforcement of high-order PDE constraints without introducing additional gradient computations.

- **PINNsFormer**: PINNsFormer employs a Transformer-based architecture to incorporate temporal context into physics-informed learning (Zhao et al., 2024). A pseudo-sequence generator converts a spatio-temporal query $(x, t)$ into a short temporal window, which is processed using multi-head attention to model temporal correlations while retaining point-wise supervision.

- **PINNMamba**: PINNMamba integrates a State Space Model (SSM) into the PINN framework to better align continuous PDE dynamics with discretely sampled training data (Xu et al., 2025a). It uses sub-sequence modeling and a contrastive alignment loss to alleviate optimization bias and improve the transmission of initial condition information.

- t-PINN in Table 1 is a simple MLP with 3 hidden layers, each with 32 neurons.

## B.2. Deep Architecture Baselines

We evaluate t-PINN against several recent state-of-the-art neural architectures on a suite of challenging PDE benchmarks. The comparison spans three backbone architectures: a standard multilayer perceptron (MLP), a modified MLP, and PirateNet. For PirateNet, *depth* corresponds to the number of adaptive residual blocks, whereas for the MLP-based architectures it denotes the number of layers. The *width* parameter specifies the number of neurons per hidden layer. Random Fourier Features (RFF) and Random Weight Factorization (RWF) are employed in PirateNet and the modified MLP, following prior work. The MLP backbone used has 3 layers with 256 neurons each.

Table 4 summarizes the architectural configurations and training hyperparameters used across all benchmark problems. To ensure comparability, we closely follow the experimental protocols and optimization settings reported in Wang et al. (2024; 2021).

*Table 4.* Architectural and training hyperparameters for the PDE benchmark problems used in the cross-architecture comparison.

| Parameter | Wave | Burgers | AC | KdV | GL |
|---|---|---|---|---|---|
| *Architecture* | | | | | |
| Depth | 3 | 3 | 3 | 3 | 3 |
| Width | 256 | 256 | 256 | 256 | 256 |
| Activation | Tanh | Tanh | Tanh | Tanh | Swish |
| RFF scale | 10.0 | 2.0 | 2.0 | 2.0 | 2.0 |
| RWF | | | $\mu = 1.0,\ \sigma = 0.1$ | | |
| *Learning rate schedule* | | | | | |
| Initial learning rate | | | $10^{-3}$ | | |
| Decay rate | | | 0.9 | | |
| Decay steps | $2 \times 10^3$ | $2 \times 10^3$ | $5 \times 10^3$ | $2 \times 10^3$ | $2 \times 10^3$ |
| Warmup steps | | | $5 \times 10^3$ | | |
| *Training* | | | | | |
| Iterations (per time window) | $10^5$ | $10^5$ | $3 \times 10^5$ | $10^5$ | $10^5$ |
| Batch size | | | 8192 | | |
| Number of time windows | 1 | 1 | 1 | 1 | 5 |
| Weighting scheme | | | GradNorm | | |
| Causal weighting tolerance | 1.0 | 1.0 | 1.0 | 1.0 | 1.0 |
| Number of chunks | 16 | 16 | 16 | 16 | 16 |

*Table 5.* t-PINN loss hyperparameters for benchmark PDEs.

| PDE | Initial degrees of freedom $\nu$ | Initial precision $\lambda$ |
|---|---|---|
| Wave | 50.0 | 0.02 |
| Burgers | 50.0 | 1.0 |
| AC | 50.0 | 1.0 |
| KdV | 50.0 | 1.0 |
| GL | 5.0 | 0.1 |
| Reaction | 5.0 | 0.002 |
| Convection | 50.0 | 0.02 |

### B.3. Loss Hyperparameters

In addition to the architectural hyperparameters, Table 5 reports the initial values of the precision parameter $\lambda$ and the degrees of freedom $\nu$ used to initialize the loss model.

To encode a prior preference for heavy-tailed residual behavior where it is most relevant, the proposed loss is applied exclusively to the interior PDE residual. All remaining constraints, including boundary conditions and initial conditions, are enforced using a standard MSE loss. This design reflects the assumption that large, non-Gaussian errors are more likely to arise from interior interfaces, multiscale structure, or model mismatch in the PDE residual, rather than from prescribed constraints.

## C. Inference and Parameter Estimation with Student-*t* Residuals

Let $\{(x_i, t_i)\}_{i=1}^N$ denote fixed collocation points and define the PDE residuals

$$r_i(\theta) := \mathcal{P}[u_\theta](x_i, t_i), \qquad \mathbf{r}(\theta) := (r_1(\theta), \ldots, r_N(\theta)).$$

We model the distribution of possible residual values via a Gaussian–Gamma hierarchy. Conditioned on latent local precisions $\boldsymbol{\eta} = (\eta_1, \ldots, \eta_N)$,

$$r_i(\theta) \mid \eta_i, \lambda \sim \mathcal{N}\big(0, (\lambda \eta_i)^{-1}\big), \qquad \eta_i \sim \text{Gam}\big(\tfrac{\nu}{2}, \tfrac{\nu}{2}\big), \tag{15}$$

where $\lambda > 0$ is a global precision parameter and $\nu > 0$ denotes the degrees of freedom. Conditional on $\boldsymbol{\eta}$, the residuals are independent with heteroskedastic variances. Marginalizing out $\boldsymbol{\eta}$ yields independent and identically distributed Student-$t$ residuals.

Marginalizing out each $\eta_i$ yields a Student-$t$ distribution for the residuals. In particular, for any $r \in \mathbb{R}$,

$$
\begin{aligned}
p(r \mid \lambda, \nu) &= \int_0^\infty \left(\frac{\lambda \eta}{2\pi}\right)^{1/2} \exp\left(-\frac{\lambda \eta}{2} r^2\right) \frac{(\nu/2)^{\nu/2}}{\Gamma(\nu/2)} \eta^{\nu/2-1} \exp\left(-\frac{\nu}{2}\eta\right) d\eta \\
&= \frac{\Gamma\big(\frac{\nu+1}{2}\big)}{\Gamma\big(\frac{\nu}{2}\big)} \left(\frac{\lambda}{\pi\nu}\right)^{1/2} \left(1 + \frac{\lambda r^2}{\nu}\right)^{-(\nu+1)/2},
\end{aligned} \tag{16}
$$

where we used the Gamma integral $\int_0^\infty z^{\alpha-1} e^{-\beta z} dz = \Gamma(\alpha)\beta^{-\alpha}$. Consequently, under the marginal model (16), the residuals $\{r_i(\theta)\}_{i=1}^N$ are independent and identically distributed according to a Student-$t$ law with $\nu$ degrees of freedom and scale $\lambda^{-1/2}$.

The marginal log-likelihood of the realized residual vector $\mathbf{r}(\theta)$ is

$$\log p\big(\mathbf{r}(\theta) \mid \lambda, \nu\big) = \sum_{i=1}^N \log p\big(r_i(\theta) \mid \lambda, \nu\big).$$

Neglecting additive constants independent of $\theta$, the corresponding negative log-likelihood defines the Student-$t$ training objective

$$\mathcal{L}_{\text{T}}(\theta; \lambda, \nu) = \sum_{i=1}^N \frac{\nu+1}{2} \log\left(1 + \frac{\lambda\, r_i(\theta)^2}{\nu}\right), \tag{17}$$

which is directly optimizable by gradient-based methods. For finite $\nu$, this loss grows sub-quadratically in the residuals and limits the influence of large violations, while in the Gaussian limit $\nu \to \infty$ it reduces to a quadratic penalty.

**Joint model, variational formulation, and EM.** To obtain a structured optimization scheme, we retain the latent-variable representation and place a Gamma prior on the global precision together with a general prior on the degrees of freedom,

$$\lambda \sim \text{Gam}(a_\lambda, b_\lambda), \qquad \nu \sim p(\nu),$$

and an optional prior $p(\theta)$ on the network parameters. The resulting joint density is

$$p(\mathbf{r}(\theta), \boldsymbol{\eta}, \theta, \lambda, \nu) = \prod_{i=1}^N p\big(r_i(\theta) \mid \eta_i, \lambda\big)\, p(\eta_i \mid \nu)\, p(\lambda)\, p(\nu)\, p(\theta). \tag{18}$$

Integrating out the latent precisions yields the marginal objective

$$\log p(\mathbf{r}(\theta), \theta, \lambda, \nu) = \log \int p(\mathbf{r}(\theta), \boldsymbol{\eta}, \theta, \lambda, \nu)\, d\boldsymbol{\eta}. \tag{19}$$

A standard variational identity gives the decomposition

$$\log p(\mathbf{r}(\theta), \theta, \lambda, \nu) = \mathcal{L}_{\text{ELBO}}(q, \theta, \lambda, \nu) + \text{KL}\big(q(\boldsymbol{\eta}) \,\big\|\, p(\boldsymbol{\eta} \mid \mathbf{r}(\theta), \theta, \lambda, \nu)\big), \tag{20}$$

where, for any distribution induced by density $q(\boldsymbol{\eta})$,

$$\mathcal{L}_{\text{ELBO}}(q, \theta, \lambda, \nu) = \mathbb{E}_{q(\boldsymbol{\eta})}[\log p(\mathbf{r}(\theta), \boldsymbol{\eta}, \theta, \lambda, \nu)] - \mathbb{E}_{q(\boldsymbol{\eta})}[\log q(\boldsymbol{\eta})].$$

Since the KL divergence is nonnegative, the ELBO is a lower bound on the marginal objective, and the bound is tight when $q(\boldsymbol{\eta}) = p(\boldsymbol{\eta} \mid \mathbf{r}(\theta), \theta, \lambda, \nu)$. It follows that

$$\log p(\mathbf{r}(\theta), \theta, \lambda, \nu) = \max_{q(\boldsymbol{\eta})} \mathcal{L}_{\mathrm{ELBO}}(q; \theta, \lambda, \nu),$$

so maximizing the marginal objective is equivalent to the nested optimization

$$\max_{\theta, \lambda, \nu} \log p(\mathbf{r}(\theta), \theta, \lambda, \nu) = \max_{\theta, \lambda, \nu} \max_{q(\boldsymbol{\eta})} \mathcal{L}_{\mathrm{ELBO}}(q; \theta, \lambda, \nu).$$

Expectation–maximization (EM) performs block coordinate ascent on this problem by alternating between updating $q(\boldsymbol{\eta})$ for fixed $(\theta, \lambda, \nu)$ (E-step) and updating $(\theta, \lambda, \nu)$ for fixed $q(\boldsymbol{\eta})$ (M-step). For the conjugate Gaussian–Gamma hierarchy (18), the E-step is available in closed form, and the resulting M-step in $\theta$ corresponds to minimizing a weighted quadratic surrogate of (17), yielding an iteratively reweighted optimization scheme.

**E-step.** For fixed $(\theta, \lambda, \nu)$, the ELBO is maximized with respect to $q(\boldsymbol{\eta})$ by choosing the exact conditional posterior,

$$q^{\star}(\boldsymbol{\eta}) = p(\boldsymbol{\eta} \mid \mathbf{r}(\theta), \theta, \lambda, \nu).$$

By conditional independence and conjugacy of the Gaussian–Gamma hierarchy (15), this posterior factorizes and each factor has Gamma density such that

$$\eta_i \mid \mathbf{r}(\theta), \theta, \lambda, \nu \ \sim \ \mathrm{Gam}\left(\tfrac{\nu+1}{2}, \tfrac{\nu + \lambda r_i(\theta)^2}{2}\right).$$

The moments required in the M-step are therefore

$$\mathbb{E}_{q^{\star}}[\eta_i] = \frac{\nu + 1}{\nu + \lambda r_i(\theta)^2}, \qquad \mathbb{E}_{q^{\star}}[\log \eta_i] = \psi\left(\tfrac{\nu+1}{2}\right) - \log\left(\tfrac{\nu + \lambda r_i(\theta)^2}{2}\right). \tag{21}$$

With this choice of $q^{\star}$, the KL divergence in (20) vanishes and the ELBO equals the marginal objective for fixed $(\theta, \lambda, \nu)$

**M-step.** Holding $q^{\star}(\boldsymbol{\eta})$ fixed, the M-step updates $(\theta, \lambda, \nu)$ by solving

$$(\theta, \lambda, \nu) \ \in \ \arg\max_{\theta, \lambda, \nu} \mathbb{E}_{q^{\star}}[\log p(\mathbf{r}(\theta), \boldsymbol{\eta}, \theta, \lambda, \nu)].$$

Due to additivity of the joint log density, the updates decouple across parameters.

**Update for $\theta$.** The $\theta$-dependent terms arise only from $p(\mathbf{r}(\theta) \mid \boldsymbol{\eta}, \lambda)$ and the prior $p(\theta)$. Up to additive constants independent of $\theta$, the objective is

$$\mathcal{L}_{\theta}(\theta) = -\frac{\lambda}{2} \sum_{i=1}^{N} \mathbb{E}_{q^{\star}}[\eta_i]\, r_i(\theta)^2 + \log p(\theta). \tag{22}$$

Defining weights

$$w_i \ := \ \mathbb{E}_{q^{\star}}[\eta_i] = \frac{\nu + 1}{\nu + \lambda r_i(\theta)^2},$$

the M-step for $\theta$ is the weighted least-squares problem

$$\theta^{(k+1)} \ \in \ \arg\min_{\theta} \left\{ \frac{\lambda}{2} \sum_{i=1}^{N} w_i\, r_i(\theta)^2 - \log p(\theta) \right\}. \tag{23}$$

Thus, EM replaces the non-quadratic Student-$t$ objective (17) by a quadratic surrogate with adaptive weights.

**Update for $\lambda$.** Collecting the $\lambda$-dependent terms in the expected complete-data log density yields

$$\mathcal{L}_{\lambda}(\lambda) = \left(\tfrac{N}{2} + a_{\lambda} - 1\right) \log \lambda - \left(b_{\lambda} + \tfrac{1}{2} \sum_{i=1}^{N} w_i\, r_i(\theta)^2\right)\lambda + \mathrm{const.} \tag{24}$$

This can be optimized analytically, and the M-step is given by

$$\lambda^{(k+1)} \ \in \ \arg\max_{\lambda > 0} \mathcal{L}_{\lambda}(\lambda) = \left\{ \frac{\tfrac{N}{2} + a_{\lambda} - 1}{b_{\lambda} + \tfrac{1}{2} \sum_{i=1}^{N} w_i\, r_i(\theta)^2} \right\}. \tag{25}$$

**Update for $\nu$.** The $\nu$-dependent part of the ELBO is

$$\mathcal{L}_\nu(\nu) = \frac{N}{2}\left[\nu\log\left(\frac{\nu}{2}\right) - \log\Gamma\left(\frac{\nu}{2}\right)\right] + \left(\frac{\nu}{2} - 1\right)\sum_{i=1}^{N}\mathbb{E}_{q^\star}[\log\eta_i] - \frac{\nu}{2}\sum_{i=1}^{N}w_i + \log p(\nu). \tag{26}$$

Since $\mathcal{L}_\nu(\nu)$ admits no closed-form maximizer, the M-step for $\nu$ is performed numerically:

$$\nu^{(k+1)} \in \arg\max_{\nu>0} \mathcal{L}_\nu(\nu).$$

Concretely, we optimize $\mathcal{L}_\nu$ by Newton iterations on the reparameterization $\nu = e^\zeta$, which enforces positivity. Writing

$$A := \sum_{i=1}^{N}\mathbb{E}_{q^\star}[\log\eta_i], \qquad B := \sum_{i=1}^{N}w_i,$$

the first and second derivatives of $\mathcal{L}_\nu$ with respect to $\nu$ are

$$\frac{\partial\mathcal{L}_\nu}{\partial\nu} = \frac{N}{2}\left[\log\left(\frac{\nu}{2}\right) + 1 - \psi\left(\frac{\nu}{2}\right)\right] + \tfrac{1}{2}A - \tfrac{1}{2}B + \frac{\partial}{\partial\nu}\log p(\nu), \tag{27}$$

$$\frac{\partial^2\mathcal{L}_\nu}{\partial\nu^2} = \frac{N}{2}\left[\frac{1}{\nu} - \frac{1}{2}\psi_1\left(\frac{\nu}{2}\right)\right] + \frac{\partial^2}{\partial\nu^2}\log p(\nu), \tag{28}$$

where $\psi(\cdot)$ and $\psi_1(\cdot)$ denote the digamma and trigamma functions, respectively.

Newton updates are performed in the unconstrained variable $\zeta = \log\nu$. By the chain rule,

$$\frac{d}{d\zeta}\mathcal{L}_\nu(e^\zeta) = \nu\frac{\partial\mathcal{L}_\nu}{\partial\nu}, \qquad \frac{d^2}{d\zeta^2}\mathcal{L}_\nu(e^\zeta) = \nu^2\frac{\partial^2\mathcal{L}_\nu}{\partial\nu^2} + \nu\frac{\partial\mathcal{L}_\nu}{\partial\nu},$$

and the Newton step is given by

$$\zeta \leftarrow \zeta - \frac{\frac{d}{d\zeta}\mathcal{L}_\nu(e^\zeta)}{\frac{d^2}{d\zeta^2}\mathcal{L}_\nu(e^\zeta)}, \qquad \nu \leftarrow e^\zeta.$$

In practice, one or two Newton iterations per EM step are sufficient. For numerical stability, $\nu$ is restricted to a bounded interval $[\nu_{\min}, \nu_{\max}]$.

---

**Algorithm 2** EM Training for PINNs with Student-$t$ Residuals

---

1: **Input:** Collocation points $\{(x_i, t_i)\}_{i=1}^N$, priors $(a_\lambda, b_\lambda)$, $p(\nu)$, optional prior $p(\theta)$
2: Initialize $\theta^{(0)}$, set $\lambda^{(0)} \leftarrow \lambda_0$ and $\nu^{(0)} \leftarrow \nu_0$.
3: **repeat**
4:     Compute residuals $r_i \leftarrow r_i(\theta^{(k)}) := \mathcal{P}[u_{\theta^{(k)}}](x_i, t_i)$ for $i = 1, \dots, N$.
5:     **E-step:** compute posterior moments under $q^{(k+1)}(\boldsymbol{\eta}) = p(\boldsymbol{\eta} \mid \mathbf{r}(\theta^{(k)}), \theta^{(k)}, \lambda^{(k)}, \nu^{(k)})$

$$w_i^{(k+1)} \leftarrow \mathbb{E}[\eta_i] = \frac{\nu^{(k)} + 1}{\nu^{(k)} + \lambda^{(k)} r_i^2}, \qquad \ell_i^{(k+1)} \leftarrow \mathbb{E}[\log \eta_i] = \psi\left(\frac{\nu^{(k)}+1}{2}\right) - \log\left(\frac{\nu^{(k)}+\lambda^{(k)} r_i^2}{2}\right).$$

6:     **M-step for $\theta$:** update network parameters by (approximately) solving

$$\theta^{(k+1)} \leftarrow \arg\min_\theta \left\{ \frac{\lambda^{(k)}}{2} \sum_{i=1}^N w_i^{(k+1)} r_i(\theta)^2 - \log p(\theta) \right\},$$

    using gradient-based optimization.
7:     **M-step for $\lambda$:** update global precision by maximizing the ELBO in closed form

$$\lambda^{(k+1)} \leftarrow \frac{\frac{N}{2} + a_\lambda - 1}{b_\lambda + \frac{1}{2} \sum_{i=1}^N w_i^{(k+1)} r_i(\theta^{(k+1)})^2}.$$

8:     **M-step for $\nu$:** update degrees of freedom by numerically maximizing $\mathcal{L}_\nu(\nu)$ using one or two Newton iterations on $\zeta = \log \nu$, and optionally clip $\nu$ to $[\nu_{\min}, \nu_{\max}]$.
9:     $k \leftarrow k + 1$.
10: **until** $\mathcal{L}_{\mathrm{ELBO}}$ converged
11: **Output:** $(\theta^{(k)}, \lambda^{(k)}, \nu^{(k)})$ and weights $\{w_i^{(k)}\}_{i=1}^N$.

---

For convenience, we collect all update steps derived above in Algorithm 2, which provides a fully explicit specification of Algorithm 1 without introducing any algorithmic modifications.

## D. Proofs of Statements in Section 5

This appendix collects proofs of all theoretical statements referenced in Section 5 (Training Dynamics). For clarity, we separate results about (i) stochastic-gradient training dynamics under the Student-$t$ loss and (ii) the surrogate/majorization interpretation underlying the EM-style updates.

### D.1. Stochastic-Gradient Training Dynamics

**Proposition 5.2** (Bounded score) *Let $\nu > 0$ and $\lambda > 0$, and consider the score function*

$$\vartheta(r) := \ell'(r) = (\nu + 1) \frac{\lambda r}{\nu + \lambda r^2}, \quad r \in \mathbb{R}.$$

*Then*

$$|\vartheta(r)| \le \frac{\nu + 1}{2} \sqrt{\frac{\lambda}{\nu}} \quad \text{for all } r \in \mathbb{R}.$$

*Moreover, the bound is attained at $r = \pm\sqrt{\nu/\lambda}$.*

*Proof.* The function $\vartheta$ is odd, hence $|\vartheta|$ is even. It therefore suffices to maximize $\vartheta(r)$ over $r \ge 0$.

For $r > 0$, differentiation gives

$$\vartheta'(r) = \frac{(\nu + 1)\lambda(\nu - \lambda r^2)}{(\nu + \lambda r^2)^2}.$$

The unique critical point satisfies $\nu - \lambda r^2 = 0$, i.e. $r = \sqrt{\nu/\lambda}$. Since $\vartheta'(r) > 0$ for $0 < r < \sqrt{\nu/\lambda}$ and $\vartheta'(r) < 0$ for $r > \sqrt{\nu/\lambda}$, this point is the global maximizer on $[0, \infty)$. Furthermore, $\vartheta(0) = 0$ and $\vartheta(r) \to 0$ as $r \to \infty$.

Evaluating at the maximizer yields

$$\vartheta\left(\sqrt{\frac{\nu}{\lambda}}\right) = \frac{\nu+1}{2}\sqrt{\frac{\lambda}{\nu}},$$

which proves the claim. Equality occurs at $r = \pm\sqrt{\nu/\lambda}$. $\square$

**Lemma D.1** (Convergence Statement). *Let*

$$\mathcal{L}_{\mathrm{T}}(\theta) = \frac{1}{N}\sum_{i=1}^{N}\ell\big(r_i(\theta)\big), \qquad \ell(r) = \frac{\nu+1}{2}\log\left(1 + \frac{\lambda r^2}{\nu}\right),$$

*and consider stochastic gradient descent iterates*

$$\theta_{k+1} = \theta_k - \alpha_k g_k,$$

*where $g_k$ is a stochastic estimator of $\nabla\mathcal{L}_{\mathrm{T}}(\theta_k)$. Assume that:*

1. *the iterates $\{\theta_k\}$ remain almost surely in a compact set $K \subset \mathbb{R}^d$,*

2. *$u_\theta$ has smooth activation functions and the residual operator is smooth, so that $\theta \mapsto r_i(\theta)$ is continuously differentiable,*

3. *the step sizes satisfy the Robbins–Monro conditions*

$$\sum_{k=0}^{\infty}\alpha_k = \infty, \qquad \sum_{k=0}^{\infty}\alpha_k^2 < \infty,$$

4. *(unbiasedness and bounded variance on $K$) for all $k$,*

$$\mathbb{E}[g_k \mid \theta_k] = \nabla\mathcal{L}_{\mathrm{T}}(\theta_k), \qquad \mathbb{E}\big[\|g_k - \nabla\mathcal{L}_{\mathrm{T}}(\theta_k)\|^2 \mid \theta_k\big] \leq \sigma^2 \quad \textit{almost surely},$$

   *for some constant $\sigma^2 < \infty$.*

*Then*

$$\lim_{k\to\infty}\|\nabla\mathcal{L}_{\mathrm{T}}(\theta_k)\| = 0 \quad \textit{almost surely}.$$

*Proof.* By Assumption (2), each $r_i(\theta)$ is continuously differentiable, hence so is $\mathcal{L}_{\mathrm{T}}$. Moreover, since the iterates lie almost surely in the compact set $K$ (Assumption (1)), the gradient $\nabla\mathcal{L}_{\mathrm{T}}$ is Lipschitz continuous on $K$.

The objective is bounded from below since $\ell(r) \geq 0$ for all $r$, hence $\inf_{\theta\in\mathbb{R}^d}\mathcal{L}_{\mathrm{T}}(\theta) \geq 0$.

Assumption (4) provides an unbiased stochastic gradient estimator with bounded conditional variance along the iterates. Together with Lipschitz continuity of $\nabla\mathcal{L}_{\mathrm{T}}$ on $K$ and the Robbins–Monro step size conditions (Assumption (3)), the almost sure stationarity result for SGD applies. In particular, the SGD corollary of Li & Milzarek (2022) (Corollary 3.1 therein) yields

$$\|\nabla\mathcal{L}_{\mathrm{T}}(\theta_k)\| \to 0 \quad \textit{almost surely}.$$

$\square$

**Remark.**

1) Note that the underlying stochastic-approximation results in Li & Milzarek (2022) are stated with conditional expectations given the full history of the algorithm. For readability, we write $\mathbb{E}[\cdot \mid \theta_k]$. This simplification is valid under standard minibatch sampling schemes, for which the stochastic gradient is unbiased and has bounded conditional variance given the current iterate $\theta_k$.

2) Assumption (4) can be verified for the Student-$t$ objective as follows. By Proposition 5.2, the score function $\vartheta(r) = \ell'(r)$ satisfies $|\vartheta(r)| \leq C_\vartheta$ for some constant $C_\vartheta > 0$. Since $\nabla_\theta r_i(\theta)$ is continuous on the compact set $K$, it is bounded: $\|\nabla_\theta r_i(\theta)\| \leq C_r$ for all $\theta \in K$ and all $i$. Therefore,

$$\|\vartheta(r_i(\theta_k)) \nabla_\theta r_i(\theta_k)\| \leq C_\vartheta C_r \quad \text{for all } i, \text{ almost surely.}$$

For a minibatch gradient with fixed size $|\mathcal{B}_k|$,

$$\|g_k\| = \left\| \frac{1}{|\mathcal{B}_k|} \sum_{i \in \mathcal{B}_k} \vartheta(r_i(\theta_k)) \nabla_\theta r_i(\theta_k) \right\| \leq \frac{1}{|\mathcal{B}_k|} \sum_{i \in \mathcal{B}_k} C_\vartheta C_r = C_\vartheta C_r,$$

and similarly $\|\nabla \mathcal{L}_\mathrm{T}(\theta_k)\| \leq C_\vartheta C_r$. Since $(a - b)^2 \leq 2a^2 + 2b^2$, we have

$$\mathbb{E}\big[\|g_k - \nabla \mathcal{L}_\mathrm{T}(\theta_k)\|^2 \mid \theta_k\big] \leq 2(C_\vartheta C_r)^2,$$

which verifies Assumption (4) with $\sigma^2 = 2(C_\vartheta C_r)^2$.

## D.2. Surrogate Majorization and Descent for the Student-$t$ Objective

**Lemma 5.3** Variational Student-$t$ objective) *Let $\nu, \lambda > 0$ and define the Student-$t$ penalty*

$$\ell(r) := \frac{\nu + 1}{2} \log\left(1 + \frac{\lambda r^2}{\nu}\right), \qquad r \in \mathbb{R}.$$

*Then $\ell$ admits the variational representation*

$$\ell(r) = \min_{w > 0} \left\{ \frac{\lambda}{2} w r^2 + \frac{\nu}{2} w - \frac{\nu + 1}{2} \log w \right\} + \kappa(\nu, \lambda), \tag{29}$$

*where $\kappa(\nu, \lambda)$ is a constant independent of $r$ and $w$. The minimizer is unique and given by*

$$w(r) = \frac{\nu + 1}{\nu + \lambda r^2}.$$

*Consequently, for any fixed $w > 0$, the right-hand side of (29) is a quadratic upper bound on $\ell(r)$ as a function of $r$, and it is tight at $r$ when $w = w(r)$.*

*Proof.* Fix $r \in \mathbb{R}$ and define, for $w > 0$,

$$F(w; r) := \frac{\lambda}{2} w r^2 + \frac{\nu}{2} w - \frac{\nu + 1}{2} \log w.$$

Since

$$\frac{\partial^2 F}{\partial w^2}(w; r) = \frac{\nu + 1}{2w^2} > 0,$$

the function $w \mapsto F(w; r)$ is strictly convex on $(0, \infty)$ and thus admits a unique minimizer. Differentiating and setting the derivative to zero yields

$$\frac{\partial F}{\partial w}(w; r) = \frac{\lambda}{2} r^2 + \frac{\nu}{2} - \frac{\nu + 1}{2} \frac{1}{w} = 0 \quad \Longrightarrow \quad w(r) = \frac{\nu + 1}{\nu + \lambda r^2}.$$

It remains to verify that substituting $w(r)$ recovers $\ell(r)$ up to an additive constant independent of $r$. Substituting $w(r)$ into $F$ gives

$$\begin{aligned}
F(w(r); r) &= \frac{\lambda}{2} w(r) r^2 + \frac{\nu}{2} w(r) - \frac{\nu + 1}{2} \log w(r) \\
&= \frac{1}{2}(\nu + \lambda r^2) w(r) - \frac{\nu + 1}{2} \log w(r) \\
&= \frac{\nu + 1}{2} - \frac{\nu + 1}{2} \log\left(\frac{\nu + 1}{\nu + \lambda r^2}\right) \\
&= \frac{\nu + 1}{2} \log(\nu + \lambda r^2) + \underbrace{\left(\frac{\nu + 1}{2} - \frac{\nu + 1}{2} \log(\nu + 1)\right)}_{\text{independent of } r}.
\end{aligned}$$

On the other hand,

$$\ell(r) = \frac{\nu+1}{2}\log(\nu + \lambda r^2) - \frac{\nu+1}{2}\log\nu.$$

Therefore,

$$\ell(r) = F(w(r); r) + \kappa(\nu, \lambda),$$

with

$$\kappa(\nu, \lambda) = -\frac{\nu+1}{2}\log\nu - \frac{\nu+1}{2} + \frac{\nu+1}{2}\log(\nu+1),$$

which is independent of $r$ and $w$. This establishes (29).

Finally, since (29) expresses $\ell(r)$ as the pointwise minimum over $w > 0$ of functions that are quadratic in $r$ (plus terms independent of $r$), fixing $w$ yields a quadratic upper bound on $\ell(r)$, and tightness at $r$ holds by construction when $w = w(r)$. $\qquad\square$

**Theorem 5.4** (Quadratic majorization and descent) *Let $\nu, \lambda > 0$ and define the Student-t objective*

$$\mathcal{L}_{\mathrm{T}}(\theta) := \sum_{i=1}^{N} \ell(r_i(\theta)), \qquad \ell(r) = \frac{\nu+1}{2}\log\left(1 + \frac{\lambda r^2}{\nu}\right).$$

*For an iterate $\theta^{(t)}$, define weights*

$$w_i^{(t)} := \frac{\nu+1}{\nu + \lambda r_i(\theta^{(t)})^2},$$

*and the quadratic surrogate*

$$Q_t(\theta) := \frac{\lambda}{2}\sum_{i=1}^{N} w_i^{(t)} r_i(\theta)^2.$$

*Define also the constant*

$$C_t := \sum_{i=1}^{N}\left(\frac{\nu}{2}w_i^{(t)} - \frac{\nu+1}{2}\log w_i^{(t)}\right) + N\,\kappa(\nu, \lambda), \tag{30}$$

*where $\kappa(\nu, \lambda)$ is the additive constant from Lemma 5.3 (it is independent of $\theta$ and $w$). Then for all $\theta$,*

$$\mathcal{L}_{\mathrm{T}}(\theta) \leq Q_t(\theta) + C_t, \tag{31}$$

*and equality holds at $\theta = \theta^{(t)}$. Consequently, any update $\theta^{(t+1)}$ satisfying*

$$Q_t(\theta^{(t+1)}) \leq Q_t(\theta^{(t)})$$

*also satisfies*

$$\mathcal{L}_{\mathrm{T}}(\theta^{(t+1)}) \leq \mathcal{L}_{\mathrm{T}}(\theta^{(t)}).$$

*Proof.* By Lemma 5.3, for each $i$ and all $r \in \mathbb{R}$,

$$\ell(r) = \min_{w>0}\left\{\frac{\lambda}{2}wr^2 + \frac{\nu}{2}w - \frac{\nu+1}{2}\log w\right\} + \kappa(\nu, \lambda). \tag{32}$$

In particular, for any fixed $w > 0$, evaluating the right-hand side of (32) at that $w$ yields an upper bound:

$$\ell(r) \leq \frac{\lambda}{2}wr^2 + \frac{\nu}{2}w - \frac{\nu+1}{2}\log w + \kappa(\nu, \lambda). \tag{33}$$

Applying (33) with $r = r_i(\theta)$ and $w = w_i^{(t)}$ and summing over $i = 1, \ldots, N$ yields, for all $\theta$,

$$\mathcal{L}_{\mathrm{T}}(\theta) = \sum_{i=1}^{N}\ell(r_i(\theta)) \leq \frac{\lambda}{2}\sum_{i=1}^{N} w_i^{(t)} r_i(\theta)^2 + \sum_{i=1}^{N}\left\{\frac{\nu}{2}w_i^{(t)} - \frac{\nu+1}{2}\log w_i^{(t)}\right\} + N\,\kappa(\nu, \lambda),$$

which is exactly (31) with the definition (30).

To show tightness at $\theta = \theta^{(t)}$, again by Lemma 5.3, the minimizer of (32) for $r = r_i(\theta^{(t)})$ is

$$w(r_i(\theta^{(t)})) = \frac{\nu + 1}{\nu + \lambda r_i(\theta^{(t)})^2} = w_i^{(t)}.$$

Therefore the upper bound (33) is tight at $r = r_i(\theta^{(t)})$ when $w = w_i^{(t)}$, i.e.

$$\ell\big(r_i(\theta^{(t)})\big) = \frac{\lambda}{2} w_i^{(t)} r_i(\theta^{(t)})^2 + \frac{\nu}{2} w_i^{(t)} - \frac{\nu + 1}{2} \log w_i^{(t)} + \kappa(\nu, \lambda).$$

Summing over $i$ yields equality in (31) at $\theta = \theta^{(t)}$.

Finally, using (31) at $\theta^{(t+1)}$ and tightness at $\theta^{(t)}$ gives

$$\mathcal{L}_{\mathrm{T}}(\theta^{(t+1)}) \le Q_t(\theta^{(t+1)}) + C_t \le Q_t(\theta^{(t)}) + C_t = \mathcal{L}_{\mathrm{T}}(\theta^{(t)}),$$

which proves monotone descent. $\square$

# E. Additional Plots on Tail Statistics of Residuals

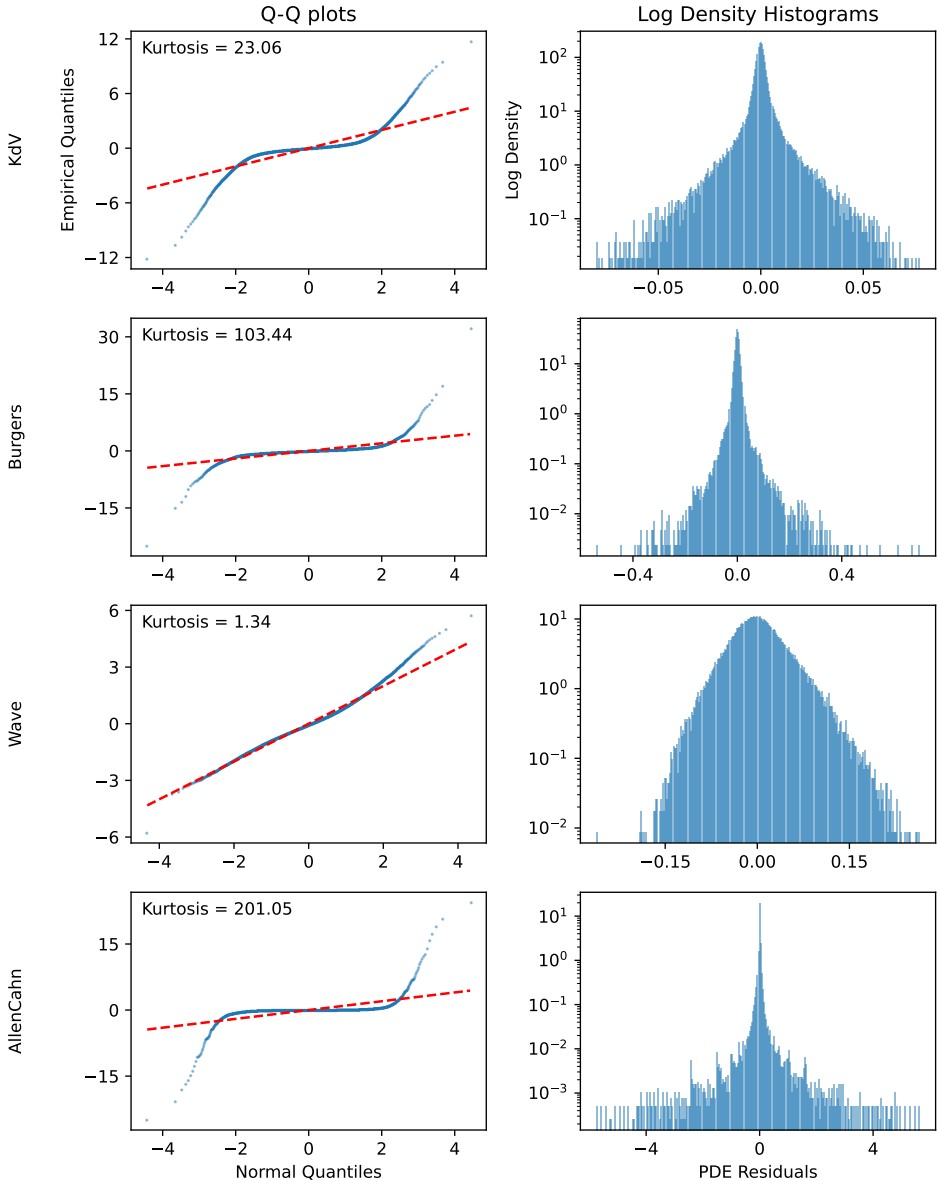

*Figure 3.* Comparison of residual distributions for different PDEs after training. Each row corresponds to a distinct PDE. Left: Normal Q–Q plot of standardized residuals, showing pronounced deviations from Gaussian quantiles and extreme excess kurtosis for non-linear PDEs. Right: Log-density of residuals, revealing a sharp central peak and slow tail decay, an indication of large PDE residuals occurring far more frequently than implied by a Gaussian noise assumption.

# F. Analysis of Heavy-tailedness of PDE Residuals

We present preliminary analysis of the heavy-tailed behavior of PDE residuals in PINNs. As a representative example, we consider the one-dimensional viscous Burgers equation on a space–time domain $\Omega \subset \mathbb{R}^2$,

$$u_t(x,t) + u(x,t)\,u_x(x,t) - \nu u_{xx}(x,t) = 0.$$

Let $u_\theta : \Omega \to \mathbb{R}$ denote a neural network approximation of the solution. Following the standard PINN formulation, $u_\theta$ is trained by minimizing an MSE loss from the PDE residual evaluated at randomly sampled collocation points $\{(x_p, t_p)\}_{p=1}^{N_r} \subset \Omega$,

$$\mathcal{L}_r(\theta) = \frac{1}{N_r} \sum_{p=1}^{N_r} |\partial_t u_\theta(x_p, t_p) + u_\theta(x_p, t_p)\, \partial_x u_\theta(x_p, t_p) - \nu\, \partial_{xx} u_\theta(x_p, t_p)|^2 \,,$$

with boundary and initial condition losses omitted for simplicity.

We analyze this problem in the small-initialization regime, where network parameters are initialized with small i.i.d. Gaussian weights and the network width is sufficiently large to admit a Gaussian process approximation near initialization. In this regime, the network output and its derivatives are jointly Gaussian at randomly sampled collocation points. Under these assumptions, the nonlinear convection term $u(x,t)\,u_x(x,t)$ induces multiplicative interactions between Gaussian quantities, which leads to heavy-tailed PDE residuals even arbitrarily close to initialization.

**Definition F.1** (Kurtosis). Let $Z$ be a real-valued random variable with finite mean $\mu = \mathbb{E}[Z]$, variance $\sigma^2 = \mathbb{E}\big[(Z - \mu)^2\big]$, and finite fourth moment. The *kurtosis* of $Z$ is defined as

$$\mathrm{kurt}(Z) \;\coloneqq\; \frac{\mathbb{E}\big[(Z - \mu)^4\big]}{\big(\mathbb{E}[(Z - \mu)^2]\big)^2}.$$

A Gaussian random variable has $\mathrm{kurt}(Z) = 3$, and the *excess kurtosis* is defined as $\mathrm{kurt}(Z) - 3$. $Z$ is called super-Gaussian (hence has heavier tails than Gaussian) when $\mathrm{kurt}(Z) > 3$.

**Lemma F.2.** *Let $(X, Y)$ be jointly Gaussian, centered random variables with $\mathrm{Var}(X) > 0$ and $\mathrm{Var}(Y) > 0$. Then:*

1. *for any $\alpha, \beta \in \mathbb{R}$, the linear combination $L = \alpha X + \beta Y$ is Gaussian and $\mathrm{kurt}(L) = 3$;*

2. *if, in addition, $X$ and $Y$ are independent and $N = XY$, then $\mathrm{kurt}(N) = 9$.*

*Proof.* For 1, $L$ is a linear map applied to a jointly Gaussian vector and is therefore Gaussian. For a centered Gaussian random variable $G$ with variance $\sigma^2$, $\mathbb{E}[G^4] = 3\sigma^4$, which yields $\mathrm{kurt}(L) = 3$ by Definition F.1.

For 2, independence implies $\mathbb{E}[X^2 Y^2] = \mathbb{E}[X^2]\mathbb{E}[Y^2]$ and $\mathbb{E}[X^4 Y^4] = \mathbb{E}[X^4]\mathbb{E}[Y^4]$. Since $X$ and $Y$ are Gaussian,

$$\mathbb{E}[X^4] = 3\big(\mathbb{E}[X^2]\big)^2, \qquad \mathbb{E}[Y^4] = 3\big(\mathbb{E}[Y^2]\big)^2.$$

Let $\sigma_X^2 = \mathbb{E}[X^2]$ and $\sigma_Y^2 = \mathbb{E}[Y^2]$. For $N = XY$,

$$\mathbb{E}[N^2] = \sigma_X^2 \sigma_Y^2, \qquad \mathbb{E}[N^4] = 9\sigma_X^4 \sigma_Y^4,$$

and therefore $\mathrm{kurt}(N) = 9$. $\qquad\square$

*Remark* F.3. The assumption that $X$ and $Y$ are independent is made for analytical clarity; allowing dependence affects constants but does not change the conclusion of super-Gaussianity.

Consider the residual operator

$$\mathcal{R}[f](x,t) = \partial_t f(x,t) + f(x,t)\,\partial_x f(x,t) - \nu\,\partial_{xx} f(x,t).$$

Let $(X, T) \sim P$ be a random collocation point in $\Omega$, independent of the network parameters, and define the induced random variables

$$U := u_\theta(X, T), \quad U_t := \partial_t u_\theta(X, T), \quad U_x := \partial_x u_\theta(X, T), \quad U_{xx} := \partial_{xx} u_\theta(X, T),$$

together with the random residual

$$R := \mathcal{R}[u_\theta](X, T) = U_t + U\,U_x - \nu U_{xx}.$$

**Assumption F.4.** The network $u_\theta$ is initialized with i.i.d. Gaussian weights and biases of variance $\varepsilon^2$ with $\varepsilon = o(1)$, and the network width is sufficiently large so that:

1. $(U, U_x, U_t, U_{xx})$ is jointly Gaussian and centered;

2. $U$ and $U_x$ are independent with nonzero variance;

3. the linear combination $A := U_t - \nu U_{xx}$ is independent of $(U, U_x)$.

**Theorem F.5.** *Under Assumption F.4, the Burgers residual $R$ is super-Gaussian in the sense of Definition F.1. More precisely, defining*

$$A := U_t - \nu U_{xx}, \qquad V := U\, U_x,$$

*and letting $\sigma_A^2 = \mathrm{Var}(A)$ and $\sigma_V^2 = \mathrm{Var}(V)$, one has*

$$\mathrm{kurt}(R) = \frac{3\sigma_A^4 + 6\sigma_A^2 \sigma_V^2 + 9\sigma_V^4}{(\sigma_A^2 + \sigma_V^2)^2} > 3.$$

*Proof.* By Assumption F.4(1), $(U, U_x, U_t, U_{xx})$ is jointly Gaussian. Since differentiation and linear combinations are linear operations, $A = U_t - \nu U_{xx}$ is Gaussian; by Lemma F.2(1), $\mathrm{kurt}(A) = 3$.

The nonlinear convection term corresponds to $V = U\, U_x$. By Assumption F.4(2) and Lemma F.2(2), $\mathrm{kurt}(V) = 9$, i.e. $\mathbb{E}[V^4] = 9\sigma_V^4$. By Assumption F.4(3), $A$ and $V$ are independent.

Writing $R = A + V$, independence implies

$$\mathbb{E}[R^2] = \sigma_A^2 + \sigma_V^2,$$

and expanding $(A + V)^4$ yields

$$\mathbb{E}[R^4] = \mathbb{E}[A^4] + 6\,\mathbb{E}[A^2]\mathbb{E}[V^2] + \mathbb{E}[V^4].$$

Substituting $\mathbb{E}[A^4] = 3\sigma_A^4$ and $\mathbb{E}[V^4] = 9\sigma_V^4$ gives the stated formula for $\mathrm{kurt}(R)$, which is strictly larger than 3 whenever $\sigma_V^2 > 0$. $\qquad\square$

*Remark* F.6 (On the independence assumption). Assumption F.4(3) is made solely for analytical convenience. In the infinite-width limit, the network $u_\theta$ converges to a smooth Gaussian process, which is typically stationary in the spatial variable $x$. In this case, $U$ and $U_{xx}$ at the same spatial point are generally correlated (indeed, for stationary kernels one has $\mathrm{Cov}(U, U_{xx}) = -\mathrm{Var}(U_x) \neq 0$), so strict independence need not hold. But this strengthens the super-Gaussianity. If one retains the exact covariance structure and evaluates moments using Isserlis' theorem, the fourth moment of the nonlinear term $U\, U_x$ is *increased* relative to the independent case. Consequently, allowing dependence between $A = U_t - \nu U_{xx}$ and $(U, U_x)$ can only strengthen the super-Gaussian character of the residual $R$.

## G. Ablation on the Hyperparameters of the t-PINN Objective

### G.1. Robustness and degrees-of-freedom Ablation.

Table 3 evaluates the impact of the EM procedure described in Algorithm 1 on the one-dimensional convection equation. In the Gaussian baseline, the E-step is omitted and all residuals are assigned unit weights $w_i \equiv 1$, resulting in an unweighted MSE objective in the M-step. Introducing the E-step of Algorithm 1 yields adaptive weights $w_i$ and corresponds to minimizing a weighted MSE in the M-step.

When the degrees-of-freedom parameter $\nu$ is fixed, weights adapt to residuals (in E-step) but with a fixed tail-heaviness schedule.. Allowing $\nu$ to be updated in the M-step further adapts the weighting through successive EM iterations, leading to lower error. In all cases, the precision parameter $\lambda$ is regularized regularized by the Gamma prior introduced in Algorithm 1, whose parameters scale with the batch size. This results in an informative prior, hence the parameter remains effectively constant during training. Results are reported as mean $\pm$ standard deviation over five random initializations using the same MLP backbone.

## G.2. Initialization

We examine the sensitivity of t-PINN to the initialization of the Student-$t$ residual parameters $\lambda$ and $\nu$ on the Allen–Cahn equation. All models use an MLP backbone trained for $10^5$ iterations. Table 6 reports the relative $\ell_2$ error across a grid of initialization values spanning several orders of magnitude in both parameters.

The results exhibit a clear and structured dependence on $(\lambda, \nu)$. Accurate and stable training is obtained for moderate values of both parameters, corresponding to a regime in which the residual model remains heavy-tailed while retaining sufficient curvature to support effective optimization. In contrast, initializations with very large $\lambda$, or with extreme combinations of small $\nu$ and large $\lambda$, consistently lead to poor performance. This behavior suggests either excessive suppression of large residuals or numerical instability during optimization.

For fixed $\lambda$, increasing $\nu$ generally degrades performance. This trend is consistent with the Student-$t$ likelihood approaching a Gaussian model, thereby recovering known failure modes associated with MSE-based PINN training. Conversely, very small $\nu$ combined with large $\lambda$ yields unstable behavior, indicating that excessively heavy-tailed residual models can also be detrimental.

Overall, the ablation identifies a well-defined region of initialization values that yields robust and accurate training. The default initialization used throughout the paper is selected from this stable regime.

*Table 6.* Ablation on initialization of the Hyperparamters for t-PINN objective on Allen-Cahn PDE trained with an MLP backbone for $10^5$ iterations. For small values of $\lambda$ and $\nu$, t-PINN consistently performs good. For larger values of $\nu$, the performance degrades. This indicates the failure of the model to capture and downweight the extreme large values of the PDE residuals.

| $\lambda/\nu$ | 1 | 5 | 10 | 100 | 1000 |
|---|---|---|---|---|---|
| 0.01 | $8.1619 \times 10^{-3}$ | $9.0313 \times 10^{-3}$ | $9.3456 \times 10^{-3}$ | $9.5298 \times 10^{-3}$ | $1.0840 \times 10^{-2}$ |
| 0.1 | $1.8738 \times 10^{-3}$ | $6.4490 \times 10^{-3}$ | $9.1109 \times 10^{-3}$ | $8.3255 \times 10^{-3}$ | $7.2368 \times 10^{-3}$ |
| 1 | $5.6705 \times 10^{-1}$ | $8.9535 \times 10^{-3}$ | $2.3435 \times 10^{-2}$ | $1.7414 \times 10^{-2}$ | $6.4430 \times 10^{-3}$ |
| 5 | $5.1221 \times 10^{-1}$ | $5.1216 \times 10^{-1}$ | $1.5320 \times 10^{-2}$ | $5.1209 \times 10^{-1}$ | $3.6231 \times 10^{-1}$ |
| 10 | $5.1326 \times 10^{-1}$ | $5.1268 \times 10^{-1}$ | $6.2152 \times 10^{-1}$ | $5.1278 \times 10^{-1}$ | $5.1305 \times 10^{-1}$ |

## G.3. Computational Overhead

We report wall-clock training times in Table 7 to quantify the computational overhead of the proposed Student-$t$ residual model. All experiments, including the MSE baselines and $t$-PINN variants, were run using the same computational resources: a single NVIDIA A100 GPU with 40GB of VRAM. We use a simple MLP backbone as in Table 2. Across all benchmark PDEs, $t$-PINN adds only a small amount of runtime relative to the MSE baseline. The overhead ranges from less than one minute for KdV and Allen–Cahn to nine minutes for Ginzburg–Landau. This small increase is expected: While E-step consists of a closed-form latent-weight update, the M-step occasionally updates the degrees of freedom using a Newton step. Consequently, the additional computation is minor compared with the forward pass, PDE residual evaluation, automatic differentiation, and backpropagation already required in standard PINN training.

## G.4. Comparison to Robust Loss Alternatives

We compare $t$-PINN with robust loss alternatives based on $L_1$ and $L_\infty$ residual objectives. Prior work has shown that $L^p$-based PINN objectives can exhibit undesirable training dynamics, including oscillatory behavior across competing residual modes (Daw et al., 2023). Nevertheless, these objectives provide a useful comparison because they also depart from the standard MSE loss and aim to reduce sensitivity to residual imbalance. For the $L_\infty$ baseline, we follow the adversarial training formulation of Wang et al. (2022a), which optimizes the maximum residual through adversarially selected samples. All methods use the same network architecture, sampling, and optimization settings. We report the relative $L^2$ error over the full space–time domain.

The results show that fixed robust penalties do not consistently improve PINN training. The $L_1$ objective performs worse than MSE on all benchmarks, suggesting that assigning uniform influence across residual magnitudes can degrade optimization. The $L_\infty$ objective is more competitive, but remains inconsistent and underperforms $t$-PINN on every benchmark. In contrast, $t$-PINN achieves the lowest relative $L^2$ error across all tested PDEs.

These results are consistent with the score-function analysis in Sections 4 and 5. MSE induces linear, unbounded influence;

*Table 7.* Wall-clock training times. Times are reported in minutes for runs under one hour and in hours:minutes for longer runs.

| PDE | MSE | $t$-PINN | $\Delta$ time |
|---|---|---|---|
| KdV | 25.4m | 25.7m | +0.4m |
| AC | 28.7m | 29.5m | +0.8m |
| Burgers | 2h53m | 2h57m | +4m |
| Wave | 20m | 23m | +3m |
| GL | 5h46m | 5h55m | +9m |
| Convection | 10m | 12m | +2m |

*Table 8.* Comparison with robust loss alternatives. We report relative $L^2$ error over the space–time domain. Lower is better.

| PDE | $L_1$ | MSE | $L_\infty$ | $t$-PINN |
|---|---|---|---|---|
| Wave | $3.1\times10^{-1}$ | $9.0\times10^{-3}$ | $8.4\times10^{-3}$ | $\mathbf{1.6\times10^{-3}}$ |
| Burgers | $3.2\times10^{-4}$ | $1.1\times10^{-4}$ | $1.5\times10^{-4}$ | $\mathbf{6.8\times10^{-5}}$ |
| Allen–Cahn | $4.0\times10^{-1}$ | $2.1\times10^{-2}$ | $1.6\times10^{-2}$ | $\mathbf{2.7\times10^{-3}}$ |
| KdV | $1.3\times10^{-1}$ | $1.6\times10^{-1}$ | $9.3\times10^{-2}$ | $\mathbf{2.9\times10^{-2}}$ |
| Ginzburg–Landau | $4.6\times10^{-1}$ | $1.9\times10^{-2}$ | $2.3\times10^{-2}$ | $\mathbf{8.6\times10^{-3}}$ |

$L_1$ assigns uniform influence across residual magnitudes; and $L_\infty$ concentrates optimization on the largest residual, with adversarial training determining where this maximum-residual behavior is enforced. The Student-$t$ score instead has bounded influence and adaptively rescales residuals according to their magnitude. This provides a more flexible mechanism than fixed robust penalties by suppressing extreme residuals without ignoring the broader residual distribution.

## H. Additional Qualitative Plots

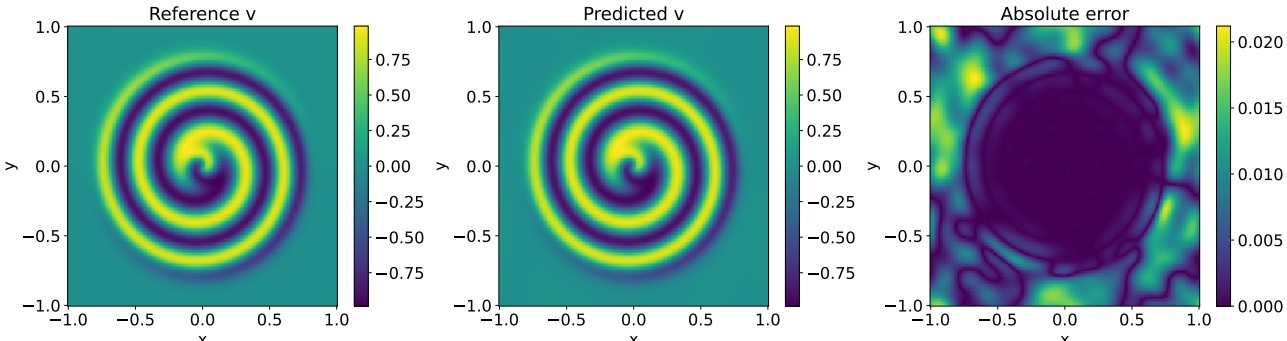

*Figure 4.* Ginzburg-Landau equation for v at the last time step: comparison between the solutions predicted by PirateNet trained with t-PINN and the reference solution.

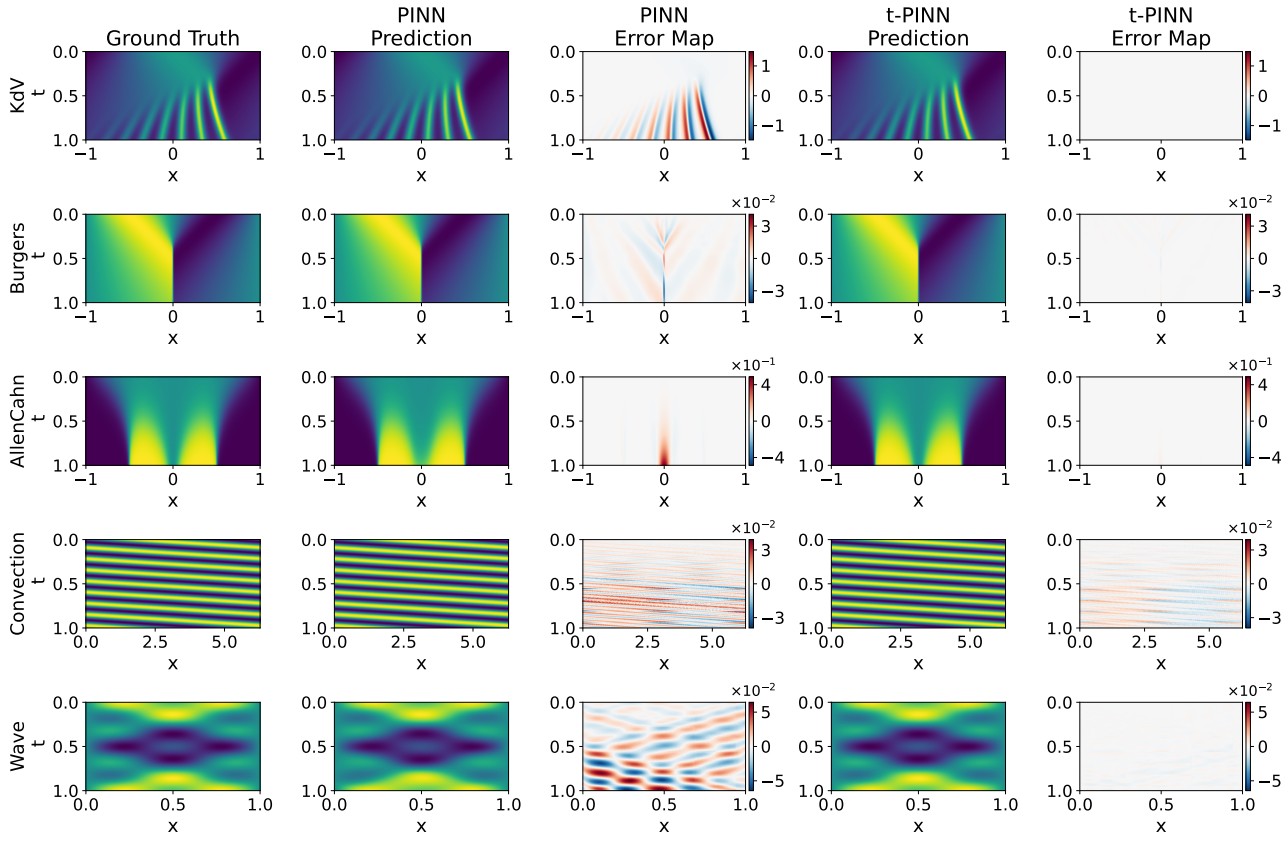

*Figure 5.* Error Maps

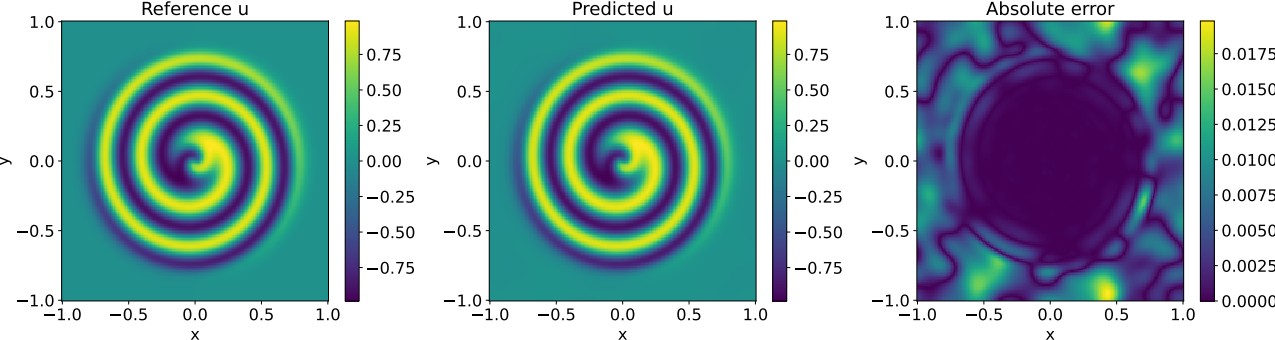

*Figure 6.* Ginzburg-Landau equation for u at the last time step: comparison between the solutions predicted by PirateNet trained with EM and the reference solution.

