# OpenReview forum: "Heavy-tailed Physics-Informed Neural Networks"
_ICML.cc/2026/Conference — ICML 2026 regular_

### Official Review · Reviewer_gKGt · 2026-03-08

**Soundness:** 3
**Presentation:** 4
**Significance:** 3
**Originality:** 4
**Overall Recommendation:** 5
**Confidence:** 3

**Summary:**

The paper explores the implicit assumption of PINNs training via MSE on residuals that residual errors are normally distributed. By analyzing the kurtosis and sample plots, the authors show that in practice these residuals have heavier-tails than Gaussians, which showcases a mismatch from theoretical assumptions. Based on this finding, the authors propose modeling PDE residuals as a tunable Student T-distribution parametrized by scalars $\lambda$ and $\nu$, which leads to an alternative loss for the maximum likelihood estimator (MLE) of the network predictions. This novel training strategy entails solving an Expectation Maximization (EM) problem, which they formalize as Algorithm 1 in the paper. Finally, the authors validate their approach, t-PINNs, in numerical experiments, where they observe improved performance across a range of benchmarks.

**Compliance With Llm Reviewing Policy:**

Affirmed.

**Final Justification:**

I think this paper proposes and interesting method with promising results, and recommend acceptance.

**Key Questions For Authors:**

(1) Have you checked the integrity/fairness of the numbers reported in Table 1 (even if they are extracted from a different paper)? I am not confident they are representative.
(2) Line 405: why is the proposed method only used for PDE residuals? It would be interesting to see a similar analysis for the other losses, even if only in the appendix.
(3) What is the difference in computational overhead for training using Algorithm 1 vs traditional MSE? It would be good to have quantitative comparisons in the main text.

Other Questions/Minor Comments
* Line 127: I think the wrong appendix is referenced here. Should it be appendix E instead of H?
* Tables: I think some of the bold to indicate best results is not accurate. For example, in Table 2 in the Allen-Cahn the best performing method is MSE-trained PirateNet, which is not bolded.

**Limitations:**

yes

**Strengths And Weaknesses:**

Strengths
* The Authors identify an implicit assumption used in the literature and show that is does not hold in practice, accurately pinpointing a common mistake in the field.
* The paper provides an actionable fix to the issue mentioned above, which results in improved performance across a range of benchmarks.
* The paper is overall well-written and easy to follow.
* The proposed method is flexible and compatible with most existing PINN frameworks.


Weaknesses
* All the PDEs reported in Table 1 are fairly simple, and may not be representative of realistic PINN use-cases. Table 2 provides more challenging benchmarks such as the KdV and GL equations, but I think as a community we should be moving towards more realistic/challenging benchmarks rather than Burger, Wave, Convection, etc.
* Also on table 1: I understand the numbers for other methods are taken from the literature (the FP64 paper), but I am not confident they are accurate/fair. Take for example the rRMSE reported for the Allen-Cahn problem: they are *three* orders of magnitude higher than what is reported in Table 2 for the same problem. I feel like the results for other methods reported here may not be reliable, and it might be more accurate to potentially re-run some of those methods using the same training pipeline as t-PINNs.


Remark About Mathematical Details:
The general ideas and results in the paper make sense to me, but due to the volume of reviews requested by ICML, I am unable to check individual derivations/proofs contained in the paper, particularly in pages 4 and 5.

---

> ### Author Rebuttal · Authors · 2026-03-30
>
> We thank the reviewer for the positive assessment and constructive feedback. We address each point below.
>
> **Q1: Fairness of Table 1 comparisons.**
> Table 1 is included to situate results relative to commonly reported benchmarks, using values from prior work (PINNFP64). Similar results have been reported across multiple independent studies on overlapping problems (Wu et al., 2024; Xu et al., 2025a).  All results in Table 2 are obtained under a unified training setup, and we consider Table 2 the primary basis for comparison as this includes much harder problems. We will clarify this distinction.
>
> **Q2: Use beyond PDE residuals.**
> We focus on PDE residuals as they arise from applying  differential operators to the network approximation, which amplifies local errors and leads to the strongest heavy-tailed behavior. Our empirical results in Figure 1, Figure 3 and the theoretical results all work in this framework. Extending the Student-$t$ formulation to boundary and initial condition losses is straightforward, and we will clarify this and include additional discussion in the appendix.
>
> **W1: Benchmark difficulty and scope.**
> We agree that moving toward more realistic PINN benchmarks is important. Our selection combines standard problems (e.g., Burgers, Wave, Convection) with more challenging nonlinear spatio-temporal PDEs (Allen–Cahn, KdV, Ginzburg–Landau) to enable comparison with prior work while also evaluating harder regimes. Improvements persist on the more challenging problems in Table 2, which we view as the primary evaluation.  We will discuss this motivation in the camera-ready version.
>
> In addition, as requested by reviewer mDYy, we conducted additional experiments on a 4D nonlinear Klein–Gordon equation using both MLP and Modified MLP architectures. We observe consistent improvements of t-PINN over MSE-based training across both architectures, as shown below. This indicates that the benefits of the proposed method persist beyond low-dimensional settings. These results will be included in the revised version.
>
> |  | MLP (MSE) | MLP (t-PINN) | Modified MLP (MSE) | Modified MLP (t-PINN) |
> |-----------|------------|--------------|---------------------|------------------------|
> | Klein–Gordon (4D) | 0.391 | **0.019** | 0.163 | **0.028** |
>
>
> **Q3: Computational overhead.**
> We will add quantitative comparisons to the main text. For reference, wall-clock times  are (reported in hours:minutes):
>
> |   | KdV   | AC    | Burgers | Wave | GL    | Convection |
> |---------|-------|-------|---------|------|-------|------------|
> | MSE     | 25.4  | 28.7m | 2h53m   | 20m  | 5h46m | 10m        |
> | t-PINN  | 25.7m | 29.5m | 2h57m   | 23m  | 5h55m | 12m        |
> | Δ time  | +0.4m | +0.8m | +4m     | +3m  | +9m   | +2m        |
>
> The overhead is small relative to total training time.
>
> **Minor Comments**:
> (1) Line 127: Yes, the reviewer is correct. This should reference Appendix E rather than Appendix H, and we will fix this in the revision.
>
> (2) The reviewer is correct. We will correct the bolding in Table 2 to accurately highlight PirateNet.
>
> ----
>
> We hope these clarifications address the reviewer’s concerns and make the contributions and scope of the paper clearer.
>
>
> ----
> References:
>
> Xu et al.(2025a,ICML): Sub-sequential physics-informed learning with state space model.
>
> Xu et al.(2025b,NeurIPS): FP64 is all you need: Rethinking failure modes in physics-informed neural networks.
>
> Wu et al.(2024,NeurIPS): Ropinn: Region optimized physics-informed neural networks.

---

> > ### Author Rebuttal · Reviewer_gKGt · 2026-04-02
> >
> > Thank you to the authors for updating their work. I maintain my positive review, recommending acceptance.

---

### Official Review · Reviewer_wejC · 2026-03-09

**Soundness:** 3
**Presentation:** 3
**Significance:** 1
**Originality:** 2
**Overall Recommendation:** 2
**Confidence:** 4

**Summary:**

Physics-informed neural networks (PINNs) use deep neural networks as ansatze
for numerical solutions to partial differential equations (PDEs). Rather than
use a conventional, e.g., piecewise polynomial, they use a neural net.  They
are flexible function families and automatic differentiation makes them easy
to train with gradient descent.  Generally this is done with a squared loss
defined at a set of collocation points, which of course corresponds to a
Gaussian likelihood on the residuals.  The paper observes that the actual
residuals when training PINNs are often heavy tailed, which leads to bad
behavior of various kinds.

So, the paper proposes to replace the Gaussian with a Student-t likelihood.
Using the standard Gaussian-Gamma hierarchical representation of the Student-t,
the authors derive an EM algorithm (Algorithm 1) that alternates between an
E-step computing closed-form per-residual weights and an M-step that optimizes
network parameters via weighted MSE (reusing existing PINN solvers), plus
closed-form updates for precision $\lambda$ and Newton updates for degrees of
freedom $\nu$.  As $\nu \to \infty$ it recovers standard MSE training.

The theoretical contributions are: a result showing nonlinear PDE operators
induce super-Gaussian residuals near initialization, a proof that
the Student-t score function is uniformly bounded (unlike MSE
where gradient contributions grow linearly), an almost-sure convergence result
for SGD on the Student-t objective, and a proof that the EM
procedure is majorization-minimization.

Experiments are performed on PINN benchmarks.

**Compliance With Llm Reviewing Policy:**

Affirmed.

**Key Questions For Authors:**

What does this method offer over iteratively reweighted least squares with
Student-t weights applied to generic nonlinear regression?  What is specific
to the PINN setting?

**Limitations:**

See weaknesses.

**Strengths And Weaknesses:**

Strengths

The paper is clearly written and easy to follow.  The observation that PINN
residuals are heavy-tailed (Figure 1) is well-presented.  The EM algorithm appears
correctly derived and the M-step reuses existing PINN solvers, which is
practical.

Weaknesses

The core contribution is replacing Gaussian likelihoods with Student-t and
then deriving EM using the Gaussian-Gamma setup. This is essentially the same as the
classical robust regression approaches in which Student-t residual models lead
to iteratively reweighted least squares (IRLS). Such formulations appear
in the robust statistics literature going back at least to Lange, Little &
Taylor (1989).

The paper mentions the IRLS connection only briefly ("can be interpreted as an
iteratively reweighted least-squares procedure") and does not engage with the
robust statistics literature where these methods originated. It is
difficult to understand what aspects of the method are novel beyond applying a
well-known robust regression framework to the PINN training objective.

Similarly, several theoretical properties discussed in the paper, such as bounded
influence of the Student-t score function and the EM/majorization interpretation,
are closely related to standard analyses of Student-t robust regression.
I find it unclear which theoretical results rely on the PINN setting specifically.

This is to say, I do not see what is specific to PINNs here beyond the
application domain.  Unless I'm missing something, this appears largely to be an
application of  classical robust regression ideas to the residual loss used in
PINN training.

More fundamentally, the paper evaluates t-PINN exclusively against other PINNs.
No comparison is made to standard numerical PDE solvers (finite elements, finite
differences, spectral methods) on any of the seven benchmarks. For the 1D
time-dependent problems considered, well-tuned classical solvers are fast and
highly accurate. The paper essentially falls directly into the trap described by
McGreivy & Hakim (2024, arXiv:2407.07218), who show that many PINN
papers claiming improvements evaluate only relative to other PINNs. Without
comparing to conventional solvers, it is difficult to assess whether PINNs
(heavy-tailed or otherwise) are a sensible approach to these problems relative to existing
alternatives.

---

> ### Author Rebuttal · Authors · 2026-03-30
>
> We thank the reviewer for the careful reading and detailed feedback. We address each point below.
>
> **What is specific to PINNs.**
> The key distinction lies in the structure of the residuals. In classical robust regression and IRLS in general, residuals measure the deviation between observed data and model predictions, i.e., $r = y - u_\theta(x)$. In contrast, PINNs are primarily *not data-driven but constraint-driven*. The residuals  take the form $r_\theta = \mathcal{P}\[u_{\theta}\](x,t)$(Section 2), where $\mathcal{P}$ is a differential operator involving higher-order derivatives which also in turn depend on the network parameters and outputs. In PINNs, this introduces additional  parameter sensitivities, which vary significantly across collocation points.
>
> **The contribution** of the paper is to identify this mismatch between residuals arising in PINNs and the Gaussian/MSE assumption  typically used in training. This is supported by empirical (Figure 1) and theoretical results (Theorem F.5) which shows that the PDE residuals exhibit heavy-tails. *The analysis of distribution families of residuals in PINN settings is thus novel*. The Student-$t$ model follows as a natural remedy.  The bounded score and majorization results are specifically used to study the SGD convergence of the method (Lemma D.1). These convergence results are also novel in PINN settings. Our empirical results on different PDE classes confirm this hypothesis.
>
>
> **Comparison to classical numerical solvers.**
> We agree that classical finite-difference, finite-element, and spectral methods are highly accurate and efficient for *forward* PDE solves. However, PINNs are additionally motivated by settings such as inverse problems, parameter identification, and data-constrained regimes, where incorporating observations and **physical constraints** in a unified framework is advantageous(Raissi et al.(2019, 2020), Almajid et al.(2022), Karniadakis et al.(2021)). Our goal is not to compete with these classical numerical methods, but to improve the optimization of PINNs.
>
> We appreciate the critique of McGreivy & Hakim (2024), which also aligns with an  increasing recognition by the SciML research community that there is  a  need for fair and standardized comparisons between PINNs and classical solvers, and recent efforts have begun to address this. As discussed by Grossman et al.(2024) and Takamoto et al.(2022), such comparisons remain challenging because the methods differ fundamentally in cost structure (training vs. solve time), evaluation mode (amortized inference vs. single-shot solution), and typical hardware usage (GPU vs. CPU).  We will add this remark in the related work.
>
> That said, following recent practice in the PINN literature, we do evaluate all methods against reference solutions obtained from high-resolution classical solvers when analytic solutions are unavailable (e.g., Table 2), consistent with prior work (e.g., Wang et al., 2025a,b and Zhongkai et al.,2024). Details of how these reference solutions are generated are provided in Appendix A.2. We will revise the manuscript to clarify this evaluation protocol and to explicitly state that our results demonstrate improvements within the PINN setting, without implying competitiveness with state-of-the-art classical solvers.
>
> **Relation to classical robust regression.**
> We agree that Student-$t$ residual modeling and its EM/IRLS formulation are classical, and we do not claim these components as novel. We will revise the paper to more explicitly acknowledge the robust statistics literature (e.g., Lange, Little & Taylor, 1989) and clarify this positioning in the related works. Our contribution is not the introduction of a new estimator, but the analysis of its role in PINN training, where the residuals and optimization dynamics differ from standard regression settings.
>
>
> We hope these clarifications address the reviewer’s concerns and make the contributions and scope of the paper clearer.
>
> ---
> References
> Raissi. et al(2019): Physics-informed neural networks: A deep learning framework for solving forward and inverse problems involving nonlinear partial differential equations.
>
> Raissi. et al (2020): Hidden fluid mechanics: Learning velocity and pressure fields from flow visualizations.
>
> Almajid et al.(2022): Prediction of porous media fluid flow using physics informed neural networks.
>
> Karniadakis et al.(2021): Physics-informed machine learning.
>
> Grossmann et al(2024): Can physics-informed neural networks beat the finite element method?
>
> Takamoto et al.(2022): PDEBENCH: An Extensive Benchmark for Scientific Machine Learning
>
> Zhongkai et al.(2024, Neurips): Pinnacle: A comprehensive benchmark of physics-informed neural networks for solving pdes
>
> Wang et al.(2024): Piratenets: Physics-informed deep learning with residual adaptive networks.
>
> Wang et al.(2024) :Understanding and mitigating gradient flow pathologies in physics-informed neural networks.

---

> > ### Author Rebuttal · Reviewer_wejC · 2026-04-03
> >
> > I have read the response from the authors and I will not be raising my score. I think without comparing to how numerical analysts actually solve these problems, this work is of limited value.  The fact that the PINN literature more generally also fails to do this does not excuse the authors from answering the most basic question: "Would someone solve a PDE with this method?"

---

> > > ### Author Response · Authors · 2026-04-07
> > >
> > > We again thank the reviewer for reading our response and for raising this important question. We believe the comment touches on two related but distinct issues: (i) whether our method (and PINNs more broadly) can solve PDEs, and (ii) how they should be evaluated.
> > >
> > > Regarding (i), the capability of PINNs to solve PDEs is well established in the literature and in practice. Our empirical results (Table 1, Table 2, Figure 2) demonstrate that the proposed method  accurately solve  PDEs in the considered settings.
> > >
> > > Regarding (ii), we agree that comparisons to classical numerical solvers are valuable. Our validation follows standard practice (which is also consistent with validation practices in numerical analysis): for problems with analytic solutions, we use them as ground truth; for more complex cases, we validate against solutions obtained from established high-resolution numerical solvers (This is discussed in Section 5 and Appendix A). Accordingly, the accuracy of our method is assessed with respect to analytic and numerical solutions, rather than relative to other PINN variants.
> > >
> > > Our work, however, focuses on the optimization problem underlying PINNs. We show, both theoretically and empirically, that PDE residuals in PINNs exhibit heavy-tailed behavior, and we introduce an approach tailored to this distribution. This leads to improved training dynamics and more accurate PDE solutions across the considered benchmarks. The goal is therefore to improve the reliability and performance of learning-based approaches.
> > >
> > > We hope this clarifies the reviewer’s concerns and resolves any remaining questions.

---

### Official Review · Reviewer_mDYy · 2026-03-12

**Soundness:** 3
**Presentation:** 3
**Significance:** 3
**Originality:** 3
**Overall Recommendation:** 5
**Confidence:** 4

**Summary:**

t-PINN replaces the standard MSE loss in PINN training with a Student-$t$ likelihood and trains it using an EM algorithm. The motivation is empirical but rigorous: inspecting PINN residuals after training on Burgers, Allen-Cahn, and KdV equations reveals heavy-tailed distributions, kurtosis values up to 201 in the case of Allen-Cahn, which is deeply inconsistent with the Gaussian assumption implicit in MSE. The Student-$t$ distribution is a natural heavy-tailed alternative because it has a Gaussian scale-mixture representation that enables EM: in the E-step, each collocation point receives a posterior precision weight $w_i = (\nu+1)/(\nu + \lambda r_i^2)$ that down-weights points with large residuals; in the M-step, the network is updated on the re-weighted loss. This is principled (not heuristic) adaptive reweighting, and the paper proves EM monotone descent (Theorem 5.4) and proves the super-Gaussianity of Burgers residuals under Gaussian initialization (Theorem F.5). Experiments on Wave, Burgers, Allen-Cahn, KdV, and Ginzburg-Landau show $5.6\times$--$7.8\times$ error reduction over standard PINN with the same MLP backbone.

**Compliance With Llm Reviewing Policy:**

Affirmed.

**Final Justification:**

My concerns have been fully addressed. The SA-PINN comparison was the most important missing piece, and the result is decisive: t-PINN outperforms SA-PINN on 4 of 5 benchmarks, with SA-PINN competitive only on the linear Wave equation, where no heavy-tail effect is expected. Wall-clock overhead is small, higher-dimensional validation is provided, and the EM heritage will be acknowledged in the revision. Will maintain the score.

**Key Questions For Authors:**

1. Self-Adaptive PINNs (SA-PINN, McClenny \& Braga-Neto, 2021, arXiv:2109.03671) trains per-point adaptive weights via gradient-based optimization, functionally similar to t-PINN's EM-derived weights. adding a comparison with SA-PINN in Tables 1--2 seems important: the key distinction is principled probabilistic derivation versus heuristic gradient-based learning, and demonstrating that t-PINN outperforms SA-PINN would substantively validate the statistical framework. if such a comparison has been run, including it would substantially change the significance assessment.

2. Theorem F.5 proves $\text{kurt}(R) > 3$ for Burgers residuals specifically. Is there evidence, theoretical or empirical, that Allen-Cahn, KdV, and Ginzburg-Landau residuals are also super-Gaussian for reasons related to the specific structure of those PDEs? the Q-Q plots in Appendix E provide empirical support, but a sketch of why different PDE types generate heavy-tailed residuals would strengthen the paper's general claim that Student-$t$ is the right model for PINN training broadly.

3. the EM procedure adds computational overhead: the E-step requires computing $w_i$ for all collocation points each iteration, and the M-step for $\nu$ requires Newton iterations. What is the wall-clock overhead relative to standard MSE-PINN for the same number of gradient steps? Is the gain in accuracy achieved in comparable wall-clock time, or does t-PINN require significantly more compute hours?

4. Student-$t$ regression via EM is a classical result (Lange, Little \& Taylor, JASA 1989). Could the paper more explicitly position t-PINN's contribution relative to this statistical heritage? The contribution is real, the kurtosis analysis of PDE residuals and the convergence proof are new, but the paper would be more credible acknowledging what it is building on.

5. all benchmarks are 1D or 2D. Presumably heavy-tailed residuals from nonlinear dynamics also occur in higher-dimensional problems, but no evidence is provided. even a brief exploration of a $d=3$ or $d=4$ PDE would help confirm that the method scales and that the heavy-tail phenomenon persists at higher dimension.

**Limitations:**

yes, adequately discussed for the most part. the paper is honest about the Gaussian initialization assumptions in Theorem F.5 (Remark F.6) and about the 1D/2D benchmark scope. The EM overhead is not quantified but is acknowledged qualitatively. SA-PINN should be acknowledged as a closely related baseline that the paper should be compared against, its absence from the limitations section is a gap.

**Strengths And Weaknesses:**

the motivation structure is the standout feature. Instead of proposing a method and then rationalizing it, the paper opens with evidence that something is wrong: Q-Q plots, kurtosis measurements for five PDEs (up to 201 for Allen-Cahn), and a theorem proving the heavy-tail property for Burgers. Diagnosis first, remedy second. That structure makes the contribution credible.

the EM derivation is solid. the hierarchical model ($w_i \sim \text{Gamma}(\nu/2, \nu/2)$, $r_i | w_i \sim \mathcal{N}(0, (\lambda w_i)^{-1})$) gives the correct closed-form Student-$t$ marginal. E-step weights $w_i^{(t)} = (\nu+1)/(\nu + \lambda r_i^2(\theta^{(t)}))$, M-step precision update, Newton iteration for $\nu$: all correct. Theorem 5.4's descent guarantee via quadratic majorization is properly proved. Textbook robust statistics applied carefully.

Theorem F.5 is a genuinely useful addition. Proving $\text{kurt}(R) > 3$ for Burgers residuals under Gaussian initialization gives theoretical grounding for why Student-$t$ is the right model, not just a post-hoc rationalization of empirical gains. Remark F.6 is honest about the independence assumption, and the argument that relaxing it would only increase kurtosis goes in the right direction.

Main concern: no comparison with self-adaptive PINNs (SA-PINN, McClenny & Braga-Neto, 2021). SA-PINN also learns per-point weights for the physics loss, via gradient-based optimization of learnable $\lambda_i$ parameters. The key distinction is principled probabilistic derivation versus heuristic gradient-based learning. Without that comparison, it's hard to tell how much of the $5.6\times$--$7.8\times$ gain comes from the Student-$t$ framework specifically versus from any adaptive weighting scheme. This comparison matters more than several included baselines.

Statistical heritage, Student-$t$ regression via EM is a classical result (Lange, Little & Taylor, JASA 1989). The contribution of t-PINN is real, but the paper presents the EM derivation as though it were novel algorithmic development. Acknowledging this lineage explicitly would make the paper more credible, not less.

Secondary concerns: all five PDEs are 1D or 2D in space, and EM overhead isn't quantified in wall-clock terms. Heavy-tail residuals presumably persist at higher dimensions, but no evidence is provided. Impossible to assess the compute story without a wall-clock comparison.

---

> ### Author Rebuttal · Authors · 2026-03-30
>
> We thank the reviewer for the constructive feedback. We address each point below and will incorporate the suggested additions in the revision.
>
> **Q1: Comparison to SA-PINN.**
> We agree that SA-PINN is an important baseline. In response to the reviewer’s suggestion, we ran SA-PINN under the same training pipeline and we will include the comparison in Table 2. For reference the  relative l2 error are:
>
> | Model   | Wave      | Burgers   | AC        | KdV       | GL        |
> |---------|-----------|-----------|-----------|-----------|-----------|
> | SA-PINN | **1.4e-3**| 4.8e-4    | 2.1e-2    | 1.2e-1    | 1.2e-2    |
> | t-PINN  | 1.6e-3    | **6.8e-5**| **2.7e-3**| **2.9e-2**| **8.6e-3**|
>
> t-PINN outperforms SA-PINN on 4/5 benchmarks, with particularly large gaps on Allen–Cahn and KdV, while SA-PINN is competitive on Wave which is linear. This shows the gains are not due to adaptive weighting alone: SA-PINN learns weights via gradient descent, whereas t-PINN derives them as posterior latent precisions under a Student-$t$ model.
>
> **Q2: Heavy-tailed residuals beyond Burgers.**
> Theorem F.5 is specific to Burgers, and this is clarified. The broader claim relies on a common structural feature: the PDE residual is composed of multiple terms (e.g., $(u_t,u,u_x, u^3)$) that  differ significantly in magnitude across the domain. Multiplicative nonlinearities further increase this heterogeneity by amplifying differences in magnitude, so that larger values grow disproportionately compared to smaller ones. As a result, the residual combines contributions with heterogeneous scales, leading to many small values and a minority of large ones. Such mixtures of scales are a standard mechanism for producing super-Gaussian (heavy-tailed) behavior, consistent with the empirical Q–Q plots in Appendix E across Allen–Cahn, KdV, and Ginzburg–Landau.
>
> **Q3: Computational overhead.**
> We agree that wall-clock cost should be reported. We will add a runtime table in the main paper of the camera-ready  version. For reference, here are the key results (reported in hours:minutes):
>
> |   | KdV   | AC    | Burgers | Wave | GL    | Convection |
> |---------|-------|-------|---------|------|-------|------------|
> | MSE     | 25.4  | 28.7m | 2h53m   | 20m  | 5h46m | 10m        |
> | t-PINN  | 25.7m | 29.5m | 2h57m   | 23m  | 5h55m | 12m        |
> | Δ time  | +0.4m | +0.8m | +4m     | +3m  | +9m   | +2m        |
>
> The overhead is small relative to total training time. The E-step is a closed-form weight update and the M-step reuses the standard optimizer.
>
> **Q4: Relation to prior work.**
> The connection to classical Student-$t$ regression via EM (e.g., Lange et al., 1989) will be made explicit in the main text.
>
> **Q5: Higher-dimensional validation.**
> We agree that higher-dimensional validation is important. In response, we conducted additional experiments on a 4D nonlinear Klein–Gordon equation using both MLP and Modified MLP architectures. We observe consistent improvements of t-PINN over MSE-based training across both architectures, as shown below. This indicates that the benefits of the proposed method persist beyond low-dimensional settings. These results will be included in the revised version.
>
> **4D Klein–Gordon (relative $\ell_2$ error, lower is better).**
>
> |  |      MLP (MSE) |    MLP (t-PINN) | Modified MLP (MSE) | Modified MLP (t-PINN) |
> |-----------|------------|--------------|---------------------|------------------------|
> | Klein–Gordon (4D) | 0.391 | **0.019** | 0.163 | **0.028** |
>
> -----
> We hope this addresses the reviewer's concerns and helps making the paper's contribution more clear

---

> > ### Author Rebuttal · Reviewer_mDYy · 2026-04-02
> >
> > The SA-PINN comparison was my primary concern, and the rebuttal addresses it directly: t-PINN outperforms SA-PINN on 4 of 5 benchmarks. The wall-clock overhead, higher-dimensional validation, and EM heritage acknowledgment are also satisfactorily addressed. Thank you for the thorough response.

---

### Official Review · Reviewer_Sdsr · 2026-03-12

**Soundness:** 4
**Presentation:** 4
**Significance:** 3
**Originality:** 3
**Overall Recommendation:** 5
**Confidence:** 4

**Summary:**

Authors show in section 2 that PINNs are trained with the underlying hypothesis that the residuals follow a zero-mean Gaussian distribution. Then, they show in Theorem F.5 and Fig. 1 that in practice this hypothesis is not correct: PINNs residuals exhibit heavy-tailed distributions and thus the loss used for training is misspecified by nature.

Starting from this observation, they propose to model the residuals of the PINNs with a Student-t distribution. In order to jointly fit both the network parameters and the distribution parameters, they propose an EM-based iterative procedure: from the current network predictions and distribution parameters, posterior expectations of the latent per-point variables eta_i are computed in closed form (E-step, eq. 9), yielding per-residual weights w_i. From these weights, the network parameters and distribution parameters are updated via a weighted MSE objective (M-step, eq. 10).

Then, in section 5 they provide two convergence guarantees: first, direct SGD on the Student-t loss is guaranteed to converge to a stationary point, second, each EM iteration is guaranteed to not increase the loss, which is expected as their method fits in the EM framework.

Finally, they show a complete experiment section where they show the use of their method benefit to several different PINNs architecture improving their results in each case (Table2). And ablation is done showing the improvement provided by optimising on $\nu$ the parameters of the prior distributions of the $\eta_i$ controlling the weights.

**Compliance With Llm Reviewing Policy:**

Affirmed.

**Ethical Review Concerns:**

/

**Final Justification:**

Authors have updated their experimentations to include comparison with other losses, and have responded to my questions on the prior distribution. I thus increase my rating to accept.

**Key Questions For Authors:**

My most important point would be to add comparison with other robust training procedure (for instance from Wang et al. (2022a)) including both the results and a comparison of the computational overhead in term of training cost.

**Limitations:**

yes

**Strengths And Weaknesses:**

Strengths :

- The paper is clearly written and follows a compelling chain of thought: starting from an empirical observation, moving to a theoretical demonstration, then proposing a mathematically grounded solution and validating it experimentally.
- The experimental section show consistent improvements across a wide range of PINN architectures and PDE benchmarks along with some ablation experiment that improve the understanding of the method.
- The paper includes helpful pedagogical elements alongside the proofs in Section 5, providing intuition for each theoretical result. In the same spirit, the parallel drawn with IRLS (Iteratively Reweighted Least Squares) nicely clarifies the approach for readers familiar with robust optimization — a citation to a standard reference on IRLS would further strengthen this.

Weaknesses

- I didn't see the Gamma prior hyperparameters ($a_\lambda$, $b_\lambda$) reported or discussed in the paper. What impact do they have on the results? Are they tuned per PDE? While Appendix G addresses sensitivity to the initialization of $\lambda$ and $\nu$, the choice of the prior itself remains opaque and could affect reproducibility.
- Several robust loss alternatives are cited in section 3 but none are included as experimental baselines. The method is shown to outperform MSE-based PINNs but it remains unclear how it compares to simpler robust alternatives. For instance, why not compare against Wang et al. (2022a) which proposes $L^p$ losses for PINNs from a similar motivation?

- Minor: Eq. 10: in practice it is impossible to compute the argmin over the network parameters, authors take a gradient step in Algorithm 1. This does not impact the convergence of the method as long as the optimization sufficiently minimizes the loss, although it could be clarified in the paper.

---

> ### Author Rebuttal · Authors · 2026-03-30
>
> We thank the reviewer for the careful reading and constructive suggestions. We address each point below and will incorporate the corresponding updates in the revision.
>
>
> **Comparison to robust loss alternatives.**
> We agree that comparisons to other robust alternatives are important. As noted in our related work, prior studies have observed that $L_p$-based PINN objectives can *introduce undesirable training dynamics, including oscillatory behavior across competing residual modes* (Daw et al., 2023). Following the reviewer’s suggestion, we nonetheless implemented $L_1$-PINN and $L_\infty$-PINN (as in Wang et al., 2022a) under the same training pipeline. Results are shown below and will be included in the revised version. All methods use the same architecture, collocation points, and optimization settings to ensure a fair comparison, and we report relative l2 across space-time domain.
>
> | Benchmark              | $L_1$          | MSE (L2)          | $L_\infty$     | t-PINN                          |
> |------------------------|---------------------|---------------------|---------------------|---------------------------------|
> | Wave                   | $3.1\times10^{-1}$  | $9.0\times10^{-3}$  | $8.4\times10^{-3}$  | $\mathbf{1.6\times10^{-3}}$      |
> | Burgers                | $3.2\times10^{-4}$  | $1.1\times10^{-4}$  | $1.5\times10^{-4}$  | $\mathbf{6.8\times10^{-5}}$      |
> | Allen--Cahn            | $4.0\times10^{-1}$  | $2.1\times10^{-2}$  | $1.6\times10^{-2}$  | $\mathbf{2.7\times10^{-3}}$      |
> | Korteweg--de Vries     | $1.3\times10^{-1}$  | $1.6\times10^{-1}$  | $9.3\times10^{-2}$  | $\mathbf{2.9\times10^{-2}}$      |
> | Ginzburg--Landau       | $4.6\times10^{-1}$  | $1.9\times10^{-2}$  | $2.3\times10^{-2}$  | $\mathbf{8.6\times10^{-3}}$      |
>
> We observe that $L_1$-PINN performs worse across all benchmarks, indicating that naive robustness can degrade optimization in this setting. $L_\infty$(trained via the adversarial procedure of Wang et al., 2022a, which is used to stabilize $L_\infty$ training) is more competitive with MSE, but remains inconsistent and underperforms t-PINN on all problems. In contrast, t-PINN achieves the best performance across all benchmarks, supporting the view that modeling the residual distribution is more effective than using a fixed robust penalty.
>
> Note: These results are consistent with our analysis of the score functions from Section 4 and 5: MSE induces linear (unbounded) influence, $L_1$ assigns uniform influence(all residuals matter equally), and $L_\infty$ concentrates on the maximum residual(the adversarial training helps to choose where this maximum should be), whereas the Student-$t$ score is bounded and adaptively scales with residual magnitude.
>
> **Gamma prior hyperparameters.**
> We use a Gamma prior with $(\alpha, \beta) = (0,0)$, fixed across all experiments and not tuned per PDE. We will report this explicitly in the revision to ensure reproducibility.
>
>
> **Clarification of Eq. (10) / Algorithm 1.**
> We agree that Eq. (10) may suggest exact minimization. In practice, the M-step is implemented via gradient-based updates, as noted in Section 5. Our convergence result does not require exact minimization, but only sufficient decrease under standard stochastic optimization assumptions (Also in  Lemma D.1). We will clarify this explicitly in the text to avoid confusion.
>
>
> **IRLS connection.**
> We thank the reviewer for this suggestion and will add a standard reference for IRLS to strengthen this connection.
>
> ----
> We hope these clarifications address the reviewer’s concerns and make the contributions and scope of the paper clearer.
>
> -----
> References:
>
> Daw et. al(2023, ICML): Mitigating propagation failures in physics-informed neural networks using retain-resample-release (R3) sampling.
>
> Wang et al.(2022, Neurips): Is l2 physics informed loss always suitable for training physics informed neural network?

---

> > ### Author Rebuttal · Reviewer_Sdsr · 2026-03-31
> >
> > My concerns have been addressed:
> >
> > - The comparison confirms the performance of the proposed approach compared to other robust methods
> > - I am reassured by the choice of prior on gamma
> > - The clarification in Eq. 10 has been addressed
> >
> > Thank you for your responses.

---

> > > ### Author Response · Authors · 2026-04-02
> > >
> > > We thank the reviewer for confirming that all concerns, have been fully resolved and for the positive assessment of our experimental results.
> > >
> > > Since the original objection has now been resolved, we would be grateful to know whether any remaining concerns prevent a higher assessment, so that we can further improve the paper accordingly.

---

### Decision · Program_Chairs · 2026-04-30

**Decision:**

Accept (regular)

**Comment:**

This paper proposes modeling the PINN residuals with  Student-t distribution, based on the heavy-tailed noise observation in practice. To jointly fit both the network and distribution parameters, they propose an EM-based iterative procedure to learn the network weights and distribution parameters.

Although the idea of robust regression itself is not novel, it is the first time it has been applied in PINNs with a reasonable motivation, and it shows advantages over the existing Gaussian-based PINNs methods.

The main issue in this paper, which leads to divergence in the reviewers' decisions, is the lack of evaluation comparing traditional numerical solvers. Reviewer wejC pointed out this issue shared with many other PINNs, even already published in other revenues (pointed out by the authors and other reviewers). The lack of comprehensive empirical evaluation along this line of papers is indeed an issue.  I might not blame the authors since they might follow the "convention" of evaluation in this area.

I thus recommend acceptance with some reservation, but suggest the authors 1) motivate the robust regression and EM in the introduction more clearly, and 2) seriously discuss the scope and action related to the comprehensive evaluation.